# DAMR: Efficient and Adaptive Context-Aware Knowledge Graph Question Answering with LLM-Guided MCTS

**Yingxu Wang[1]**[*]**Shiqi Fan[2]**[*]**Mengzhu Wang[3], Siyang Gao[4], Chao Wang[5], Nan Yin[4,6]**[†]
[1] Mohamed bin Zayed University of Artificial Intelligence    [2] Hong Kong Polytechnic University
[3] Hebei University of Technology   [4] City University of Hong Kong
[5] Ke Holdings Inc.   [6] Hong Kong Space Robotics and Energy Centre Limited
{yingxv.wang,dreamkily,chanceycn,yinnan8911}@gmail.com
siyangao@cityu.edu.hk, comp-shiqi.fan@connect.polyu.hk

## Abstract

Knowledge Graph Question Answering (KGQA) aims to interpret natural language queries and perform structured reasoning over knowledge graphs by leveraging their relational and semantic structures to retrieve accurate answers. Existing methods primarily follow either the retrieve-then-reason paradigm, which relies on Graph Neural Networks (GNNs) or heuristic rules to extract static candidate paths, or dynamic path generation strategies that employ large language models (LLMs) with prompting to jointly perform retrieval and reasoning. However, the former lacks adaptability due to static path extraction and the absence of contextual refinement, while the latter suffers from high computational costs and limited evaluation accuracy because of their dependence on fixed scoring functions and repeated LLM calls. To address these issues, this paper proposes **D**ynamically **A**daptive **M**CTS-based **R**easoning (DAMR), a novel framework that integrates LLM-guided Monte Carlo Tree Search (MCTS) with adaptive path evaluation to enable efficient and context-aware KGQA. DAMR leverages MCTS as a backbone, where an LLM-based planner selects the top-$k$ semantically relevant relations at each expansion step to effectively reduce the search space. To enhance evaluation accuracy, we introduce a lightweight Transformer-based scorer that performs context-aware plausibility estimation by jointly encoding the question and relation sequence through cross-attention, thereby capturing fine-grained semantic shifts during multi-hop reasoning. Furthermore, to mitigate the scarcity of high-quality supervision, DAMR incorporates a dynamic pseudo-path refinement mechanism that periodically generates training signals from partial paths explored during search, enabling the scorer to continually adapt to the evolving distribution of reasoning trajectories. Extensive experiments on multiple KGQA benchmarks show that DAMR significantly outperforms state-of-the-art methods.

## 1 Introduction

Knowledge Graph Question Answering (KGQA) (Dammu et al., 2025; Saxena et al., 2020; Choi et al., 2023) leverages structured relational knowledge to enable factual, interpretable, and robust reasoning. By combining the expressiveness of natural language with the precision of knowledge graphs, KGQA provides a grounded framework for complex reasoning and reliable answer generation (Liu et al., 2025; Yao et al., 2025). Recent advances integrate Large Language Models (LLMs) into KGQA to enhance semantic understanding and generalization, bridging the gap between symbolic reasoning and neural representation learning.

Existing KGQA approaches can be broadly categorized into two paradigms according to how reasoning paths are constructed: retrieve-then-reason methods and dynamic path generation strategies.

---

[*]Equal contribution.
[†]Corresponding author.

In the former, candidate paths are extracted prior to answer prediction, typically using Graph Neural Networks (GNNs) (Ma et al., 2025a; Yao et al., 2025) or rule-based heuristics (Fang et al., 2024). However, such methods exhibit limited adaptability, as GNNs fail to incorporate question-specific semantics during inference, while heuristic rules remain inherently inflexible and cannot support dynamic refinement of reasoning (Liu et al., 2025; Yao et al., 2025). In contrast, dynamic path generation strategies unify retrieval and reasoning by constructing paths dynamically during question processing. These methods either leverage LLMs to iteratively generate paths through in-context learning or Chain-of-Thought (CoT) (Sui et al., 2024; Li et al., 2024b), or employ guided search techniques such as MCTS, where paths are incrementally expanded with a path scorer (Ma et al., 2025b; Shen et al., 2025). While offering greater flexibility, such approaches suffer from substantial computational overhead due to repeated LLM calls and limited evaluation accuracy, as static scorers cannot capture the evolving semantics of reasoning paths (Chang et al., 2024; Lee et al., 2024).

This paper investigates the design of an adaptive KGQA framework to address challenges of computational inefficiency and limited path evaluation accuracy in dynamic reasoning. However, developing such a framework raises three key challenges: (1) *How to modularize reasoning to reduce LLM overuse in search?* Inefficiency in dynamic KGQA stems from repeatedly invoking LLMs for relation retrieval and reasoning in multi-hop path construction (Shen et al., 2025; Long et al., 2025b). Although methods such as CoT and MCTS enable flexible exploration, they bind LLMs to every decision step, causing high inference cost and poor scalability. The challenge is to design a modular framework that leverages LLMs efficiently, guiding search without direct involvement at each step. (2) *How to accurately evaluate evolving reasoning paths?* As multi-hop paths are incrementally constructed, their semantics evolve with each added relation and context. Yet existing methods rely on static scoring or shallow similarity metrics that fail to capture these semantic shifts (Xu et al., 2024; Sui et al., 2024). This highlights the challenge of designing a path evaluation model that adaptively captures fine-grained changes conditioned on both the question and evolving relation sequence. (3) *How to train a reliable evaluation model with limited supervision?* Accurate path ranking requires a well-calibrated scorer, yet dynamic reasoning generates many incomplete paths with only a few valid ones. This results in imbalanced, noisy supervision, especially for multi-hop questions where successful trajectories are scarce. While reinforcement learning has been explored (Ma et al., 2024a; Zhai et al., 2024), it suffers from sparse rewards and unstable optimization. The key challenge is to construct reliable learning signals from limited supervision to support adaptive scorer training.

To address these challenges, we propose **D**ynamically **A**daptive **M**CTS-based **R**easoning (DAMR), an efficient framework that integrates LLM-guided MCTS with context-aware semantic modeling for accurate and efficient KGQA. DAMR employs an MCTS backbone, where an LLM-based planner dynamically guides path expansion by selecting semantically relevant relations at each step, thereby reducing the search space and improving answer identification. For path evaluation, we introduce a lightweight Transformer-based scorer that jointly encodes the question and relation sequence via cross-attention, enabling context-sensitive plausibility estimation and capturing evolving semantics during multi-hop reasoning. To mitigate supervision scarcity, DAMR further incorporates a dynamic pseudo-path mechanism that continuously adapts the scorer during search: partial paths from MCTS rollouts are ranked by predicted plausibility and converted into pseudo-path supervision pairs, amplifying signals from promising trajectories while suppressing noise from suboptimal ones. Our contributions are summarized as follows:

- We study adaptive path reasoning in KGQA, where the key challenges lie in capturing the evolving semantics of multi-hop reasoning paths and ensuring computational efficiency during search, motivating the need for dynamic and context-aware reasoning strategies.
- We propose DAMR, a novel framework that integrates MCTS with a dynamically adapted path evaluation model, enhancing evaluation accuracy while maintaining computational efficiency.
- We conduct extensive experiments across multiple KGQA benchmarks, demonstrating that DAMR consistently outperforms state-of-the-art methods.

## 2 RELATED WORK

**Knowledge Graph Question Answering (KGQA).** KGQA aims to enhance reasoning capabilities by incorporating external knowledge graphs to answer natural language questions (Choi et al., 2023; Xu et al., 2025). Existing KGQA approaches can be broadly classified into two categories:

retrieve-then-reason and dynamic path generation. The first category extracts candidate reasoning paths using Graph Neural Networks (GNNs)(Ma et al., 2025a; Yao et al., 2025) or rule-based heuristics(Fang et al., 2024), followed by LLM-based answer generation. While GNNs learn embeddings to identify relevant paths and rule-based methods apply predefined patterns (Zhao et al., 2023; Liu et al., 2025), these approaches lack the flexibility to adapt dynamically to question-specific context during inference. In contrast, dynamic path generation methods, such as CoT prompting (Sui et al., 2024; Li et al., 2024b) and MCTS (Ma et al., 2025b; Shen et al., 2025), unify retrieval and reasoning for more flexible exploration. However, they suffer from high computational overhead due to repeated LLM calls, and static scorers often fail to adapt to evolving path semantics (Long et al., 2025b; Luo et al., 2025; Ma et al., 2024b). To address these challenges, we propose an adaptive framework that integrates symbolic search with a fine-tuned evaluation model, aiming to improve both computational efficiency and reasoning accuracy in KGQA.

**Adaptive and Self-Improving Reasoning Models.** A promising approach to developing adaptive reasoning models is to frame the process within a reinforcement learning (RL) paradigm, where an agent learns a policy to navigate a state space. Early methods such as DeepPath (Xiong et al., 2017) and MINERVA (Das et al., 2018) used RL to discover reasoning paths by rewarding the agent only when a correct answer is reached. However, this leads to the sparse rewards problem, as positive feedback arrives only after long action sequences, resulting in weak learning signals and poor exploration efficiency (Zhai et al., 2024; Chang et al., 2023). To address this challenge, an alternative is self-training via pseudo-labeling, where the model learns from its own high-confidence predictions (Lee et al., 2013; Xie et al., 2020). While commonly used in semi-supervised learning, pseudo-labeling proves especially effective in reasoning tasks with limited supervision (Wang et al., 2022; Huang et al., 2025). Instead of relying on sparse terminal rewards, we leverage intermediate search paths as dynamic pseudo-paths, offering dense and adaptive supervision. This facilitates continual refinement of the path evaluator to better capture the evolving semantics of reasoning.

## 3 METHODOLOGY

### 3.1 PROBLEM FORMULATION

We define Knowledge Graph Question Answering (KGQA) as the task of answering a natural language question by reasoning over a knowledge graph (KG). The KG is typically represented as a set of triples $\mathcal{K} = \{(e_s, r, e_o)\} \subseteq \mathcal{E} \times \mathcal{R} \times \mathcal{E}$, where $\mathcal{E}$ and $\mathcal{R}$ denote the sets of entities and binary relations. The goal of KGQA is to find a set of answers $\mathcal{A}_q \subseteq \{(e_1, r_1, e_2), (e_2, r_2, e_3) \cdots\}$ for question $q$, such that a reasoning path through the KG leads from a topic entity to the correct answer. Formally, this is often framed as mapping $q$ to an executable query program $p_q$, where $\text{LLM}(p_q|\mathcal{K}) = \mathcal{A}_q$.

### 3.2 OVERVIEW OF FRAMEWORK

In this paper, we propose a dynamically adaptive reasoning framework DAMR for KGQA, as shown in Fig. 1. DAMR comprises three components: (1) **LLM Guided Expansion.** DAMR employs MCTS to incrementally expand reasoning paths, guided by an LLM-based planner that proposes relevant relations. This reduces computational overhead and enhances efficiency in KG exploration; (2)

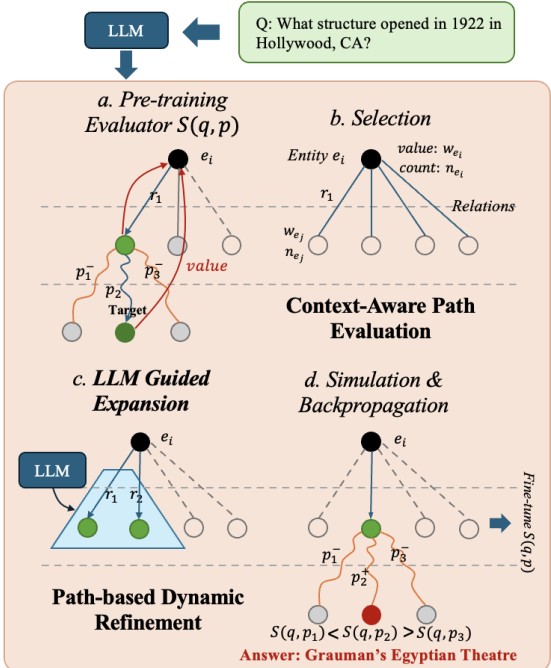

Figure 1: Overview of DAMR. The reasoning process begins with an MCTS guided by an LLM-based planner, which selects top-$k$ semantically relevant relations at each expansion step. A context-aware path evaluator scores each candidate path during simulation. To enable continual adaptation, high-confidence pseudo-paths generated during search are used to dynamically fine-tune the evaluator.

**Context-Aware Path Evaluation.** To capture the evolving semantics of reasoning paths, DAMR employs a lightweight Transformer-based scorer with cross-attention to jointly encode the question and path embeddings. This enables context-sensitive evaluation and improves the accuracy and relevance of multi-hop reasoning; (3) **Path-based Dynamic Refinement.** DAMR uses intermediate paths from MCTS as dynamic pseudo-paths to iteratively fine-tune the path evaluator, enhancing its ability to capture question-specific semantics and improving reasoning accuracy.

## 3.3 LLM GUIDED EXPANSION

A key challenge in KGQA is efficiently exploring the vast search space of multi-hop reasoning paths, especially under weak or no supervision. Existing methods often struggle to balance search efficiency and semantic relevance, resulting in either redundant exploration or missed correct paths (Long et al., 2025b; Li et al., 2024a). To address this, the LLM-Guided Expansion module employs MCTS (Kocsis & Szepesvári, 2006) as the backbone for symbolic path expansion. At each step, an LLM proposes semantically relevant relations, narrowing the search space and improving path quality, while MCTS ensures a balanced trade-off between exploration and exploitation.

Specifically, each node in the MCTS represents a reasoning state anchored at a specific entity in the KG. Given the current state, possible actions correspond to selecting an outgoing relation to extend the reasoning path. During the **Selection** phase, the search traverses from the root to a leaf node by recursively choosing children with the highest Upper Confidence Bound for Trees (UCT) score (Kocsis & Szepesvári, 2006), which is defined as:

$$UCT = \frac{w_i}{n_i} + C\sqrt{\frac{\ln N}{n_i}}, \tag{1}$$

where $w_i$ denotes the accumulated reward of node $i$, $n_i$ its visit count, $N$ the visit count of its parent, and $C$ a constant that balances exploration and exploitation. This criterion guides the search to trade off between exploring new relations and exploiting promising paths. In the **Expansion** phase, the selected leaf node is expanded by retrieving the outgoing relations of its entity $e_i$, denoted as $\mathcal{R}_{e_i} = \{r_1, r_2, \ldots, r_n\}$. To avoid exhaustive branching, we prompt an LLM with the question $q$ and candidate relations $\mathcal{R}_{e_i}$, selecting the top-$k$ relations most semantically aligned with $q$:

$$\mathcal{R}_{top-k} = \text{LLM}(q, \mathcal{R}_{e_i}). \tag{2}$$

These top-$k$ relations are then used to generate new child nodes, ensuring that expansion favors semantically meaningful directions and remains computationally efficient.

## 3.4 CONTEXT-AWARE PATH EVALUATION

While LLM-guided expansion effectively narrows the search space by selecting semantically relevant relations, it does not ensure that all expanded paths remain valid in the broader reasoning context. As the search progresses, path semantics evolve dynamically, and early promising trajectories may later become misleading or irrelevant. Prior works attempt to filter or pre-plan paths to improve relevance (Fang et al., 2024; Long et al., 2025a), but still leave gaps in capturing evolving path semantics during search. To address this, we integrate a lightweight Transformer-based path scorer into MCTS's simulation phase, employing cross-attention to jointly encode the question and current path, enabling adaptive evaluation that reflects evolving semantics.

**Context-Aware Path Evaluator.** In the **Simulation** phase, we assess the quality of candidate paths generated during MCTS rollouts. Given a question $q$ and a candidate relation path $p_r = (r_1, r_2, \ldots, r_l)$, where $p_r$ is formed by sequentially selecting relations during expansion, both $q$ and $p_r$ are first encoded using a pre-trained LLM. Let $\mathbf{z}_q \in \mathbb{R}^d$ denote the embedding of $q$ and $\mathbf{z}_{r_i} \in \mathbb{R}^d$ the embedding of relation $r_i$. To capture the sequential structure of $p_r$, we introduce a learnable positional encoding $\mathbf{e}_i^{\text{pos}}$ for each relation. The final input sequence is obtained by combining $\mathbf{z}_{r_i}$ with $\mathbf{e}_i^{\text{pos}}$ and feeding the sequence into a Transformer encoder:

$$\mathbf{E}_{p_r} = \text{Transformer}([\mathbf{z}_{r_1} + \mathbf{e}_1^{\text{pos}}, \ldots, \mathbf{z}_{r_l} + \mathbf{e}_l^{\text{pos}}]), \tag{3}$$

where $\mathbf{e}_i^{\text{pos}} = \mathbf{E}^{\text{pos}}[i]$ is the positional embedding of the $i$-th hop, drawn from a trainable matrix $\mathbf{E}^{\text{pos}} \in \mathbb{R}^{L \times d}$, where $L$ is the maximum path length and $d$ the embedding dimension. To further

incorporate question-specific information, we apply a cross-attention mechanism, allowing the encoded path representation $\mathbf{e}_{p_r}$ to attend to the question embedding $\mathbf{z}_q$:

$$\mathbf{H} = \mathbf{E}_{p_r} + \text{CrossAttn}(\mathbf{E}_{p_r}, \mathbf{z}_q), \; with \; \text{CrossAttn}(\mathbf{E}_{p_r}, \mathbf{z}_q) = \text{softmax}(\mathbf{E}_{p_r} \cdot \mathbf{z}_q^T / \sqrt{d_k}) \cdot \mathbf{z}_q. \quad (4)$$

We then apply attention pooling over the relation $\mathbf{H}$ to obtain the hidden state of the relation path:

$$\mathbf{s}_{p_r} = \sum i = 1^l \alpha_i \mathbf{h}_i, \quad \alpha = \text{Softmax}\big(\text{MLP}(\mathbf{H})\big), \quad (5)$$

where $\mathbf{h}_i$ is the hidden state of the $i$-th relation and $\alpha_i$ its learned attention weight. This pooling allows the model to emphasize informative steps along the reasoning path. The pooled path representation $\mathbf{s}_{p_r}$ is then concatenated with the question embedding $\mathbf{z}_q$, and the combined vector is passed through a multi-layer perceptron to compute the plausibility score of the question–path pair:

$$S(q, p_r) = \text{MLP}\big([\mathbf{s}_{p_r} : \mathbf{z}_q]\big). \quad (6)$$

This context-aware evaluator assigns plausibility scores to partial reasoning paths by jointly encoding the question and relation sequence, providing accurate, context-sensitive guidance for MCTS.

**Pre-training of Evaluator.** To train the context-aware evaluator, we construct supervision by sampling positive and negative paths from local subgraphs. A path is positive if it links the head entity to a correct answer within a hop limit. Negatives come from two sources: hard negatives that terminate near but miss the answer, and random negatives from walks that avoid answer entities. Each training instance is a triplet $(q, p^+, p^-)$, with sequences zero-padded and masked for efficient batch training.

The model computes a plausibility score $S(q, p)$ for each question-path pair and is optimized using the Pair-wise Ranking loss (Rendle et al., 2012) to encourage higher scores for positive paths:

$$\mathcal{L}_{\text{PR}} = -\frac{1}{M} \sum_{i=1}^{M} \log \sigma \left( S(q, p_i^+) - S(q, p_i^-) \right), \quad (7)$$

where $\sigma(\cdot)$ is the sigmoid function. This training strategy equips the evaluator with the ability to distinguish plausible reasoning paths, thereby improving the guidance signal during MCTS inference.

## 3.5 Path-based Dynamic Refinement

While LLM-guided expansion and semantic scoring improve path exploration, the static evaluator may fail to generalize to the evolving search space. To address this, we introduce a dynamic refinement mechanism that leverages high-confidence paths from MCTS rollouts as pseudo-paths. These pseudo-paths serve as supervision signals, enabling continual adaptation of the evaluator to new reasoning contexts without requiring additional labeled data.

Specifically, during **Backpropagation** phase, the plausibility score estimated by the context-aware path evaluator is propagated along the visited nodes in the MCTS tree after each simulation. For every entity $e_i$ on the simulated path, we update its visit count and aggregated value as follows:

$$n_{e_i} = n_{e_i} + 1, \quad w_{e_i} = \frac{\sum_j n_{e_j} \cdot w_{e_j}}{\sum_j n_{e_j}}, \quad (8)$$

where $n_{e_i}$ is the visit count and $w_{e_i}$ is the aggregated value of entity $e_i$. The value is computed as a weighted average over its child nodes $\{e_j\}$, and reflects the plausibility scores $w_{e_j}$ assigned during simulation. These updates refine the UCT estimates used in future selection steps, progressively biasing the search toward high-quality reasoning paths.

To construct supervision signals for fine-tuning, we dynamically sample pseudo-path pairs $(\hat{p}_i', \hat{p}_j')$ from the set of explored paths during MCTS. Instead of relying on the evaluator's predictions, we assign pseudo-labels based on empirically grounded values derived from the search process. Specifically, for entity $e_i$ along a reasoning path $p_r$, we define its search value as: $w_{e_i} = \frac{w_{p_r}}{n_{e_i}}$, where $w_{p_r}$ is the cumulative reward from all rollouts passing through $p_r$, and $n_{e_i}$ is the visite count of entity $e_i$. Given a pair of paths, we assign pseudo-labels based on their relative values:

$$\left( \hat{p}^+, \hat{p}^- \right) = \begin{cases} (p_i', p_j'), & \text{if } w_{e_i} > w_{e_j}, \\ (p_j', p_i'), & \text{otherwise.} \end{cases} \quad (9)$$

The path evaluator is then fine-tuned using the Pair-wise Ranking loss in Eq. 7, encouraging higher scores for more promising paths.

### 3.6 REASONING PROCESS

The overall reasoning process is outlined in Appendix A. The framework first initializes a path evaluator to discriminate between plausible and implausible KG paths, providing a foundation for downstream search. During dynamic MCTS, the algorithm iteratively performs selection, expansion, simulation, and backpropagation. In expansion, an LLM-based planner adaptively selects the top-$k$ relations most relevant to the question, steering the search toward semantically meaningful paths. The evaluator guides simulation by prioritizing trajectories likely to yield correct answers, while pseudo-path pairs sampled during search are periodically used for refinement. Finally, entities reached by high-scoring paths are aggregated to form the answer set.

### 3.7 THEORETICAL ANALYSIS

While DAMR dynamically refines the path evaluator during search, an important question remains: *Is this self-improving process stable and convergent?* In this part, we provide a theoretical analysis showing that DAMR's dynamic pseudo-label refinement is both statistically stable and provably convergent. The analysis focuses on three properties: (1) variance reduction of the pseudo-label estimator, (2) directional consistency of the pairwise ranking objective, and (3) global convergence of the joint refinement process.

**Lemma 1 (Stability of Pseudo-label Aggregation)** *Under the assumption that rollout samples $s_j^{(t)}$ for node $e$ are i.i.d. sub-Gaussian with variance proxy $\sigma_t^2$ and $n_e^{(t)}$ increases monotonically, then:*

$$\mathbb{E}[w_e^{(t+1)}] = V^{(t)}(e), \qquad \mathrm{Var}[w_e^{(t+1)}] \leq \sigma_t^2/n_e^{(t+1)}.$$

*As $\sigma_t^2$ decreases with $t$ (the scorer becomes more consistent) and $n_e^{(t)}$ grows, the variance of pseudo-labels exhibits a* double decay *and converges to zero.*

**Proof 1** *Sub-Gaussian aggregation implies variance reduction by $\mathrm{Var}[\bar{X}] = \sigma^2/n$. Applying this to $s_j^{(t)}$ yields the stated bound, with $\sigma_t^2$ capturing rollout dispersion. As $n_e^{(t)} \uparrow$ and $\sigma_t^2 \downarrow$, $w_e^{(t)}$ converges to its expectation $V^{(\infty)}(e)$ in $L_2$.*

This lemma guarantees that the pseudo-labels for self-supervision become statistically stable as training proceeds. Even if early rollouts are noisy, repeated exploration and averaging prevent error amplification, ensuring subsequent scorer updates rely on smooth, reliable targets.

**Lemma 2 (Directional Consistency of Scorer Updates)** *Let $(p^+, p^-)$ denote a pseudo-pair where $p^+$ has a higher aggregated value than $p^-$. If $\Pr[p^+ \succ p^-] \geq 1/2 + \gamma$ for some margin $\gamma > 0$, then the expected gradient of the pairwise loss is aligned with the gradient of the true ranking risk:*

$$\mathbb{E}[\nabla L_{\mathrm{PR}}(S)] \cdot \nabla \mathcal{R}_{\mathrm{rank}}(S) \leq -c\gamma \|\nabla \mathcal{R}_{\mathrm{rank}}(S)\|_2^2,$$

*where $c > 0$ depends on the curvature of the logistic link. This means that each update step reduces ranking error in expectation, even when pseudo-labels are noisy.*

**Proof 2** *Under the margin condition, the logistic surrogate is a proper scoring rule: the expected sign of the score difference $\Delta = S(q, p^+) - S(q, p^-)$ is positive with probability at least $\frac{1}{2} + \gamma$. Taking expectation over pairwise samples yields a negative correlation between the expected gradient of $L_{\mathrm{PR}}$ and the gradient of true ranking risk, scaled by $c\gamma > 0$ determined by logistic curvature.*

This result ensures that each scorer update reduces ranking error in expectation, rather than drifting due to noisy pseudo-labels. The scorer thus consistently learns to prefer plausible reasoning paths, stabilizing the refinement process and mitigating self-confirmation bias.

**Proposition 1 (Contraction and Fixed-point Convergence)** *Define the composite refinement map:*

$$\mathcal{G}(S^{(t)}) = \mathrm{Update}\big(S^{(t)}; w^{(t+1)}(S^{(t)})\big),$$

*where $w^{(t+1)}(S^{(t)})$ is the label map generated by MCTS aggregation and $\mathrm{Update}(\cdot)$ performs a gradient step on $\mathcal{L}_{PR}$. If both maps are Lipschitz continuous with constants $L_w$ and $L_S$ satisfying $L_S L_w < 1$, then*

$$\|\mathcal{G}(S) - \mathcal{G}(S')\|_2 \leq L_S L_w \|S - S'\|_2,$$

*and $\mathcal{G}$ is a contraction mapping. By Banach's fixed-point theorem, there exists a unique stable point $S^*$ such that $S^{(t)} \to S^*$ as $t \to \infty$.*

**Proof 3** *Since* Update *is $L_S$-Lipschitz and $w^{(t+1)}$ is $L_w$-Lipschitz in expectation, we have $\|\mathcal{G}(S) - \mathcal{G}(S')\|_2 \le L_S L_w \|S - S'\|_2$. When $L_S L_w < 1$, $\mathcal{G}$ is a contraction, implying convergence to a unique fixed point.*

This proposition provides a formal guarantee that DAMR's dynamic refinement process converges: the interaction between evolving pseudo-labels and scorer updates constitutes a contraction mapping, ensuring that the model stabilizes rather than oscillating or diverging.

Overall, the analysis reveals that DAMR's dynamic refinement achieves statistical stability, directional consistency, and convergence to a fixed equilibrium. These theoretical properties elucidate the mechanism by which DAMR mitigates feedback drift and sustains robustness when trained solely on self-generated supervision.

## 4 EXPERIMENTS

### 4.1 EXPERIMENTAL SETTINGS

**Datasets.** To evaluate the effectiveness of DAMR, we conduct experiments on two widely used KGQA benchmarks: WebQSP Talmor & Berant (2018) and CWQ Yih et al. (2016). Following prior work Sun et al. (2023); Liu et al. (2025), we uniformly sample 1,000 questions from the test sets of both datasets to evaluate the performance. More details about datasets are provided in Appendix B.

**Baselines.** We compare DAMR with a comprehensive set of baselines. These baselines include: the semantic parsing methods, e.g., KV-Mem (Miller et al., 2016), EmbedKGQA (Saxena et al., 2020), QGG (Lan & Jiang, 2020), NSM (He et al., 2021), TransferNet (Shi et al., 2021), and KGT5 (Saxena et al., 2022); the retrieval-based methods, e.g., GraftNet (Sun et al., 2018), PullNet (Sun et al., 2019), SR+NSM (Zhang et al., 2022), and SR+NSM+E2E (Zhang et al., 2022); the general LLMs, including Flan-T5-xl (Chung et al., 2024), Alpaca-7B (Taori et al., 2023), Llama3-8B (Dubey et al., 2024), Qwen2.5-7B (Team, 2024), ChatGPT (Schulman et al., 2022), and ChatGPT+CoT (Wei et al., 2022); and recent LLMs with KG methods, including UniKGQA (Jiang et al., 2022),DECAF (Yu et al., 2022), KD-CoT (Wang et al., 2023), Nutrea (Choi et al., 2023), ToG (Sun et al., 2023), RoG (Luo et al., 2023), KAPING (Baek et al., 2023), ReasoningLM (Jiang et al., 2023), FiDeLis (Sui et al., 2024), GNN-RAG (Mavromatis & Karypis, 2024), DoG (Ma et al., 2025a), DualR (Liu et al., 2025) , DP (Ma et al., 2025b), and RwT (Shen et al., 2025). The details are provided in Appendix C.

**Implementation Details.** We implement the DAMR framework using PyTorch, and all experiments are conducted on NVIDIA A100 GPUs. The LLM-based planner is implemented with GPT-4.1 (Liu et al., 2023), while question and relation embeddings are generated from Qwen3-Embedding-8B (Yang et al., 2025) with an embedding dimension of 1024. For the path evaluation module, we use a 128-dimensional embedding and employ the Adam optimizer (Kingma, 2014) with a learning rate of $1 \times 10^{-4}$ during pretraining and $1 \times 10^{-5}$ during fine-tuning. The model consists of two Transformer layers and is trained for 15 epochs in the pretraining stage and 10 epochs in the fine-tuning stage. Following (Luo et al., 2023; Yao et al., 2025; Ma et al., 2025b), we evaluate DAMR using Hits@1 and F1 score, assessing answer correctness and overall accuracy for questions with potentially multiple correct answers.

### 4.2 PERFORMANCE COMPARISON

We report the experimental results of DAMR in Table 1, benchmarking its performance against state-of-the-art baselines across KGQA datasets. From the results, we find that: (1) Semantic parsing and retrieval-based methods serve as early foundations for KGQA by extracting subgraphs and capturing structural semantics. However, embedding-based models struggle with complex relational patterns, while retrieval-based methods rely on rigid pipelines that limit generalization. In contrast, LLM with KG approaches combine the language understanding of LLMs with structured reasoning over KGs, enabling more flexible path exploration and improved adaptability to diverse, multi-hop queries. (2) General-purpose LLMs, such as ChatGPT and Llama3-8B, show basic reasoning ability but often perform worse than methods that combine LLMs with KGs in KGQA tasks. This is mainly because

| Type | Methods | WebQSP | | CWQ | |
|---|---|---|---|---|---|
| | | Hits@1 | F1 | Hits@1 | F1 |
| Semantic Parsing | KV-Mem (Miller et al., 2016) | 46.7 | 34.5 | 18.4 | 15.7 |
| | EmbedKGQA (Saxena et al., 2020) | 66.6 | - | 45.9 | - |
| | QGG (Lan & Jiang, 2020) | 73.0 | 73.8 | 36.9 | 37.4 |
| | NSM (He et al., 2021) | 68.7 | 62.8 | 47.6 | 42.4 |
| | TransferNet (Shi et al., 2021) | 71.4 | - | 48.6 | - |
| | KGT5 (Saxena et al., 2022) | 56.1 | - | 36.5 | - |
| Retrieval | GraftNet (Sun et al., 2018) | 66.4 | 60.4 | 36.8 | 32.7 |
| | PullNet (Sun et al., 2019) | 68.1 | - | 45.9 | - |
| | SR+NSM (Zhang et al., 2022) | 68.9 | 64.1 | 50.2 | 47.1 |
| | SR+NSM+E2E (Zhang et al., 2022) | 69.5 | 64.1 | 49.3 | 46.3 |
| LLMs | Alpaca-7B Taori et al. (2023) | 51.8 | - | 27.4 | - |
| | Llama3-8B (Dubey et al., 2024) | 30.3 | 25.7 | 30.5 | 27.8 |
| | Qwen2.5-7B (Team, 2024) | 28.4 | 23.7 | 25.9 | 24.1 |
| | ChatGPT (Schulman et al., 2022) | 66.8 | - | 39.9 | - |
| | ChatGPT+CoT (Wei et al., 2022) | 75.6 | - | 48.9 | - |
| LLMs+KGs | Nutrea (Choi et al., 2023) | 77.4 | 72.7 | 53.6 | 49.5 |
| | ToG (Sun et al., 2023) | 81.9 | 76.0 | 68.5 | 60.2 |
| | RoG (Luo et al., 2023) | 80.8 | 70.8 | 57.8 | 56.2 |
| | KAPING (Baek et al., 2023) | 72.4 | 65.1 | 53.4 | 50.3 |
| | ReasoningLM (Jiang et al., 2023) | 78.5 | 71.0 | 69.0 | 64.0 |
| | FiDeLis (Sui et al., 2024) | 84.3 | 78.3 | 71.5 | 64.3 |
| | GNN-RAG (Mavromatis & Karypis, 2024) | 82.8 | 73.5 | 62.8 | 60.4 |
| | DoG (Ma et al., 2025a) | 65.4 | 55.6 | 41.0 | 46.4 |
| | DualR (Liu et al., 2025) | 81.5 | 71.6 | 65.3 | 62.1 |
| | DP (Ma et al., 2025b) | 87.5 | 81.4 | 75.8 | 69.4 |
| | RwT (Shen et al., 2025) | 87.0 | 79.7 | 72.4 | 66.7 |
| | **DAMR** | **94.0** | **81.7** | **78.0** | **75.1** |

Table 1: Performance comparison (%) on WebQSP and CWQ datasets. **Bold** indicate the best results.

they are not grounded in domain-specific knowledge, making them more likely to produce incorrect or made-up answers. (3) DAMR consistently outperforms all baselines across both datasets, showcasing its strong reasoning capability. This superior performance is driven by its integration of an LLM-based planner, which selectively retrieves relevant relations to reduce noise and guide the search toward high-quality reasoning paths, and a path evaluation model that is dynamically fine-tuned during search to capture semantic differences among candidate paths and accurately rank those most likely to yield correct answers.

## 4.3 EFFICIENCY ANALYSIS

As shown in Table 2, DAMR achieves substantial improvements in computational efficiency. It reduces the average number of LLM calls to 7.1 on WebQSP and 16.8 on CWQ, with corresponding token usage of 3,931 and 9,266. These correspond to reductions of over 50% in LLM calls and 75% in token consumption relative to the strongest baseline. This efficiency is achieved by invoking the LLM only during the expansion phase of MCTS to select the top-$k$ semantically relevant relations, which effectively narrows the search space and avoids redundant reasoning steps that lead to unnecessary computational overhead. During simulation, the context-aware path evaluator efficiently assesses candidate paths based on question-path alignment without requiring any further LLM interaction or model inference. These design choices reduce both the frequency and verbosity of LLM usage while maintaining strong reasoning performance, making DAMR more efficient, scalable, and practically deployable than previous work.

## 4.4 ABLATION STUDY

We conduct ablation studies to assess the contributions of key components in DAMR: (1) DAMR w/o PE: removing the path evaluation module; (2) DAMR w/o FT: disabling fine-tuning of the evaluator; and (3) DAMR w/ GPT-4.1: replacing the context-aware evaluator with a general LLM.

| Method | WebQSP | | CWQ | |
|---|---|---|---|---|
| | #Tokens | #Calls | #Tokens | #Calls |
| DoG | 22,538 | 30.9 | 37,741 | 58.1 |
| ToG | 16,372 | 23.2 | 26,183 | 41.9 |
| RwT | 10,680 | 15.1 | 17,885 | 28.6 |
| DAMR | **3,931** | **7.1** | **9,266** | **16.8** |

Table 2: Statistics of average number of LLM calls and token consumption per question on WebQSP and CWQ datasets.

| Method | WebQSP | | CWQ | |
|---|---|---|---|---|
| | Hits@1 | F1 | Hits@1 | F1 |
| DAMR w/o PE | 91.2 | 78.2 | 74.3 | 72.1 |
| DAMR w/o FT | 91.9 | 80.1 | 75.1 | 73.0 |
| DAMR w/ GPT 4.1 | 92.5 | 79.8 | 74.9 | 72.4 |
| DAMR | **94.0** | **81.7** | **78.0** | **75.1** |

Table 3: The results of ablation studies on the WebQSP and CWQ datasets. **Bold** results indicate the best performance.

Experimental results are summarized in Table 3. From the results, we find that: (1) Removing the path evaluation module (DAMR w/o PE) causes a noticeable performance drop on both datasets, underscoring its critical role in guiding the search process. Without this component, the model cannot effectively assess or rank candidate paths, leading to suboptimal reasoning and reduced accuracy. (2) Compared to DAMR w/o FT, the proposed DAMR consistently achieves superior results on both datasets, highlighting the importance of the finetuning mechanism in the path evaluation module. This mechanism enables the model to adapt to the evolving distribution of explored paths, improving its ability to distinguish plausible from implausible reasoning trajectories. (3) Replacing the context-aware path evaluation module with general LLMs yields degraded performance, confirming the advantage of our fine-tuned path scorer. By capturing fine-grained semantic distinctions among candidate paths, it provides more accurate evaluation signals, enhancing overall search effectiveness.

## 4.5 SENSITIVITY ANALYSIS

We conduct a sensitivity analysis to assess the impact of two key hyperparameters in DAMR: the number of selected relations $k$ and the maximum reasoning path length $L$. The parameter $k$ controls how many relations are proposed by the LLM-based planner at each step, while $L$ determines the number of reasoning hops allowed during path construction.

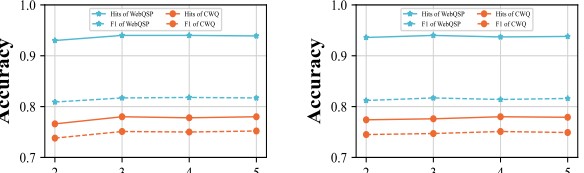

(a) Number of selected relations $k$    (b) Reasoning path length $L$

Figure 2: Sensitivity analysis of hyperparameter on the WebQSP and CWQ datasets.

Figure 2 illustrates how $k$ and $L$ affect the performance of DAMR on the WebQSP and CWQ datasets. We vary $k$ and $L$ within the range of $\{2, 3, 4, 5\}$. From the results, we observe that: (1) As shown in Figure 2(a), increasing $k$ initially leads to performance gains, which then stabilize before experiencing a slight decline. While larger $k$ values encourage broader relational exploration, they may also introduce irrelevant candidates and increased computational cost. Conversely, smaller $k$ restrict the diversity of the search. To balance these trade-offs, we select a moderate $k = 3$ as the default setting. (2) As shown in Figure 2(b), on the WebQSP dataset, performance improves from $L = 2$ to 3, then fluctuates between $L = 3$ and 5, suggesting limited gains beyond three hops. In contrast, performance on the CWQ dataset steadily increases up to $L = 4$ before slightly declining at $L = 5$, reflecting its need for deeper reasoning due to more complex questions. Balancing effectiveness and efficiency across both datasets, we set $L = 4$ as the default path length in all experiments. More results are provided in Appendix E.4.

## 4.6 IMPACT OF DIFFERENT LLMS

To evaluate the impact of different LLM-based planners within the DAMR framework, we compare several backbones including Llama2-13B (Roque, 2025), Qwen3-14B (Team, 2024), GPT 4.1 mini, and GPT 4.1, as shown in Table 4. Across both datasets, stronger LLMs consis-

| Method | WebQSP | | CWQ | |
|---|---|---|---|---|
| | Hits@1 | F1 | Hits@1 | F1 |
| DAMR (Llama2-13B) | 91.0 | 76.7 | 73.9 | 69.5 |
| DAMR (Qwen3-14B) | 91.5 | 77.8 | 74.4 | 70.1 |
| DAMR (GPT 4.1-mini) | 93.1 | 80.6 | 76.1 | 72.7 |
| DAMR (GPT 4.1) | **94.0** | **81.7** | **78.0** | **75.1** |

Table 4: Performance of DAMR using different LLM-based planners as backbones on different datasets.

| Question | Where does greek language come from? |
|---|---|
| **Answer** | Attic Group |
| **Llama-2-13B** | The Greek language originates from the Proto-Indo-European language family, which spread to Greece around 2000 BCE. It evolved into the ancient Greek language, influenced by Mycenaean and Aeolic dialects, and eventually developed into Modern Greek. |
| **Qwen-3-14B** | The Greek language originates from the Proto-Hellenic branch of the Indo-European language family. It evolved from the Mycenaean Greek of the second millennium BCE and became the foundation for Classical Greek, spoken in ancient Greece. |
| **GPT 4.1** | The Greek language originates from the Indo-European language family. It developed in the southern Balkans around the 3rd millennium BCE and is the oldest recorded living language, with written records dating back to the 15th century BCE (Linear B script). |
| **GPT 4.1-mini** | The Greek language originates from the Hellenic branch of the Indo-European language family. It developed in the region of Greece and the Aegean around the 3rd millennium BCE, evolving through ancient, medieval, and modern stages. |
| **DAMR** | `Path 1:` Entity (id: 120026) → base.rosetta.languoid.parent → Attic Group. |

(a) One example from the WebQSP dataset.

| Question | What structure opened in 1922 in Hollywood, CA? |
|---|---|
| **Answer** | Grauman's Egyptian Theatre |
| **Llama-2-13B** | The Grauman's Chinese Theatre, a historic movie palace, opened on November 18, 1922, in Hollywood, CA. It was built by Sid Grauman and Charles E. Toberman and has since become a famous landmark and tourist attraction. |
| **Qwen-3-14B** | The TCL Chinese Theatre, originally known as the Chinese Grand Theatre, opened in 1922 in Hollywood, CA. It is a historic movie theater renowned for its unique architecture and celebrity handprints and footprints. |
| **GPT 4.1** | The Hollywood Bowl, an iconic amphitheater in Hollywood, California, opened in 1922. It is renowned for its distinctive band shell and has hosted numerous concerts and events, becoming a significant cultural landmark in the area. |
| **GPT 4.1-mini** | The Hollywood Bowl, an iconic amphitheater in Hollywood, California, opened in 1922 and has since been a renowned venue for music performances and cultural events. |
| **DAMR** | `Path 1:` Entity (id: 83076) → location.location.events → time.event.locations → travel.travel_destination.tourist_attractions → Grauman's Egyptian Theatre. `Path 2:` Entity (id: 83076) → travel.travel_destination.tourist_attractions → Grauman's Egyptian Theatre. |

(b) One example from the CWQ dataset.

Figure 3: Case studies of DAMR on the WebQSP and CWQ datasets. We highlight the correct answers in Red and the wrong answers in Blue.

tently yield higher F1 and Hits scores, with GPT 4.1 achieving the best performance across all metrics. These results highlight the pivotal role of advanced LLMs in guiding relation selection and reasoning path expansion, where improved fluency, contextual awareness, and semantic precision translate into more accurate and faithful reasoning trajectories. Notably, the consistent performance gap between smaller and larger backbones illustrates the sensitivity of KGQA systems to the planner's reasoning capability, reinforcing the necessity of leveraging high-capacity models when available. Overall, these findings emphasize the importance of backbone selection and further validate the design of DAMR, which capitalizes on powerful LLMs to achieve robust, generalizable, and effective multi-hop reasoning.

## 4.7 CASE STUDY

Figure 3 presents detailed case studies on the WebQSP and CWQ datasets comparing the reasoning process of DAMR with four representative LLMs: Llama-2-13B, Qwen-3-14B, GPT 4.1-mini, and GPT 4.1. In Figure 3a, the baseline LLMs generate fluent and seemingly plausible answers such as "Proto-Indo-European" or "Proto-Hellenic" when asked about the origin of the Greek language, yet fail to identify the correct answer `Attic Group`. In contrast, DAMR accurately predicts the correct entity by explicitly traversing the *base.rosetta.languoid.parent* relation within the knowledge graph. In Figure 3b, none of the baseline models are able to identify the structure that opened in Hollywood in 1922, while the proposed DAMR successfully locates `Grauman's Egyptian Theatre` by explicitly following relevant relation paths in the knowledge graph and integrating complementary reasoning trajectories. These examples collectively highlight the strength of DAMR in grounding its reasoning in the knowledge graph and explicitly modeling multi-step reasoning paths, enabling it to provide accurate and faithful answers to complex, ontology-specific questions that general-purpose LLMs often struggle to resolve.

## 5 CONCLUSION

In this work, we presented DAMR, a dynamically adaptive MCTS-based framework for knowledge graph question answering that integrates an LLM-guided planner, a context-aware path evaluator, and a dynamic pseudo-path refinement mechanism. By narrowing the search space, reducing redundant LLM calls, and continually adapting path evaluation, DAMR achieves efficient and accurate multi-hop reasoning. Extensive experiments on WebQSP and CWQ demonstrate that DAMR consistently surpasses state-of-the-art baselines in both performance and efficiency, while ablation and case studies highlight its interpretability and robustness. These results establish DAMR as a scalable solution for real-world KGQA and open avenues for extending adaptive symbolic–neural reasoning to broader domains such as scientific discovery and recommendation.

ETHICS STATEMENT

This work complies with the ICLR Code of Ethics. Our study on KGQA relies solely on publicly available datasets, which contain no personal or sensitive information. The proposed DAMR framework is developed to enhance the efficiency and reliability of multi-hop reasoning while avoiding harm or discrimination. All experimental settings are transparently reported, with fair comparisons to prior methods, and proper credit is given to related work. The contributions are intended exclusively for scientific and educational purposes, supporting the advancement of AI research.

ACKNOWLEDGMENT

This research was supported in part by Hong Kong Research Grants Council (Grant 11217925) and National Science Foundation of China (Grant 72371214).

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

# A   ALGORITHM

---

**Algorithm 1** Dynamic MCTS-based KGQA with Path Model Pretraining and Online Refinement

---

**Input:** Question $q$, knowledge graph $\mathcal{G} = (\mathcal{E}, \mathcal{R}, \mathcal{T})$, number of selected relations $k$, MCTS iterations $N$, length of reasoning path $L$

**Output:** Answer set $\mathcal{A}$

1: / Stage 1: Path Evaluation Model Pre-training /
2: Construct reasoning path pairs $(q, p^+, p^-)$ from $\mathcal{G}$
3: Initialize path evaluation model $S(q, \cdot; \Theta)$
4: **for** each batch in pretraining data **do**
5:     Update $S(q, \cdot; \Theta)$ by minimizing the Pair-wise Ranking loss in Eq.(4)
6: **end for**
7: / Stage 2: Dynamic MCTS Reasoning /
8: **for** $i = 1$ to $N$ **do**
9:     **Selection:** Traverse the tree from root to a leaf node by selecting child nodes according to the UCT criterion in Eq.(1)
10:     **Expansion:**
11:     i. At the selected node, enumerate all candidate relations from current entities and use the LLM-based planner to select the top-$k$ most relevant relations
12:     ii. Expand a new child node for each selected relation
13:     **Simulation:** For each expanded node, perform a rollout by sequentially selecting relations (guided by the path evaluation model) up to $L$ hops or until a correct answer is reached
14:     **Backpropagation:** Update the value ($w_i$) and visit ($n_i$) statistics along the traversed path from the leaf node back to the root using the score from the simulation, as per Eq.(6)
15:     **Path Evaluation Model Fine-tuning:** Generate the explored pseudo-path pairs $(\hat{p}^+, \hat{p}^-)$ via Eq.(5) and fine-tune the path evaluation model $S(q, \cdot; \Theta)$ based on Pair-wise Ranking loss in Eq.(4)
16: **end for**
17: / Stage 3: Answer Extraction /
18: Collect entities reached by high-scoring reasoning paths as $\mathcal{A}$
19: **return** $\mathcal{A}$

---

# B   DATASETS

## B.1   DATASET DESCRIPTION

| Datasets | #Train | #Valid | #Test |
|----------|--------|--------|-------|
| WebQSP   | 2,848  | 250    | 1,639 |
| CWQ      | 27,639 | 3,519  | 3,531 |

Table 5: Statistics of KGQA datasets.

We conduct extensive experiments on two widely used multi-hop Knowledge Graph Question Answering (KGQA) benchmarks: WebQSP Talmor & Berant (2018) and CWQ Yih et al. (2016). The statistics of these two benchmarks can be found in Table 5, and their details are shown as follows:

- The WebQuestionsSP (WebQSP) dataset is a widely adopted benchmark for evaluating single-hop and simple multi-hop KGQA Yih et al. (2016). It consists of 4,837 natural language questions annotated with corresponding SPARQL queries over the Freebase knowledge graph. The dataset is partitioned into 2,848 training, 250 validation, and 1,639 test instances.

- The ComplexWebQuestions (CWQ) dataset is a challenging benchmark designed for multi-hop KGQA Talmor & Berant (2018). It comprises 34,689 questions derived from WebQuestionsSP, reformulated to include more complex and compositional queries. Each

question typically requires multi-step reasoning over the Freebase knowledge graph, often involving conjunctions, comparatives, or nested logical structures. The dataset is divided into 27,639 training, 3,519 validation, and 3,531 test examples.

## B.2 DATA PROCESSING

Following prior work Shen et al. (2025); Long et al. (2025b); Ma et al. (2025b), we preprocess the datasets by constructing localized subgraphs centered around each question entity to reduce the size of the search space. Specifically, for each question in WebQSP Yih et al. (2016) and CWQ Talmor & Berant (2018), we extract a subgraph from the Freebase knowledge graph by including all triples within a predefined number of hops from the topic entity. This approach preserves the essential context required for multi-hop reasoning while significantly improving computational efficiency.

## C  BASELINES

In this part, we introduce the details of the compared baselines as follows:

- **Semantic Parsing Methods.** We compare our DAMR with six semantic parsing methods:
  - **KV-Mem**: KV-Mem Miller et al. (2016) introduce a neural architecture that stores facts as key-value pairs and enables question answering by attending over memory slots, directly retrieving relevant information to infer answers.
  - **EmbedKGQA**: EmbedKGQA Saxena et al. (2020) enhances multi-hop question answering over knowledge graphs by leveraging pretrained knowledge base embeddings, enabling the model to reason over entity and relation representations without explicit path enumeration during answer prediction.
  - **QGG**: QGG Lan & Jiang (2020) generates query graphs to answer multi-hop complex questions over knowledge bases, formulating question answering as query graph prediction and enabling structured reasoning through graph matching and path ranking mechanisms.
  - **NSM**: NSM He et al. (2021) enhances multi-hop KBQA by leveraging intermediate supervision signals, decomposing questions into reasoning steps, and training a neural state machine to sequentially predict relations and entities for accurate path-based reasoning.
  - **TransferNet**: TransferNet Shi et al. (2021) proposes a transparent framework for multi-hop QA over relational graphs by transferring question semantics to relation paths through interpretable path ranking and structured reasoning, enabling effective and explainable answer prediction.
  - **KGT5**: KGT5 Saxena et al. (2022) formulates knowledge graph completion and question answering as unified sequence-to-sequence tasks, leveraging pre-trained language models to jointly encode input queries and generate answer entities or triples in a flexible and end-to-end manner.

- **Retrieval-Based Methods.** We compare our DAMR with four retrieval-based methods:
  - **GraftNet**: GraftNet Sun et al. (2018) proposes an early fusion framework that jointly encodes knowledge base facts and supporting text by constructing a heterogeneous graph, enabling effective reasoning through graph convolutional networks for open-domain question answering.
  - **PullNet**: PullNet Sun et al. (2019) introduces an iterative retrieval mechanism that expands a query-specific subgraph by pulling relevant facts from both knowledge bases and text, enabling joint reasoning over heterogeneous evidence for open-domain question answering.
  - **SR+NSM**: SR+NSM Zhang et al. (2022) enhances multi-hop KBQA by first retrieving a question-relevant subgraph and then performing symbolic reasoning over it using Neural Symbolic Machines, improving efficiency and accuracy through constrained and focused logical inference.

- **SR+NSM+E2E**: SR+NSM+E2E Zhang et al. (2022) extends SR+NSM by enabling end-to-end training that jointly optimizes subgraph retrieval and reasoning. This integration enhances model coherence and allows better alignment between retrieved subgraphs and final answer prediction.

- **General Large Language Models (LLMs).** We compare our DAMR with six general LLMs:

  - **Flan-T5-xl**: Flan-T5-xl Chung et al. (2024) is an instruction-finetuned variant of the T5 model, trained on a diverse collection of tasks with natural language instructions. By leveraging large-scale instruction tuning, it improves zero-shot and few-shot performance across diverse NLP benchmarks.

  - **Alpaca-7B**: Alpaca-7B Taori et al. (2023) is an instruction-following language model fine-tuned from LLaMA-7B using self-instruct techniques. It demonstrates strong zero-shot and few-shot performance by aligning with human instructions across various NLP tasks.

  - **Llama3-8B**: Llama3-8B Dubey et al. (2024) is part of the LLaMA 3 family of models, designed for improved instruction following, reasoning, and code generation. Pre-trained on a high-quality corpus and fine-tuned with supervised signals, it achieves strong performance across diverse benchmarks.

  - **Qwen2.5-7B**: Qwen2.5-7B Team (2024) is a 7B-parameter instruction-tuned language model developed by Alibaba, optimized for tasks such as reasoning, code generation, and dialogue. It supports multi-turn conversation and demonstrates competitive performance on standard benchmarks.

  - **ChatGPT**: ChatGPT Schulman et al. (2022) is a conversational AI developed by OpenAI, based on the GPT architecture. It is designed to understand natural language, engage in dialogue, answer questions, and assist with a wide range of tasks across domains.

  - **ChatGPT+CoT**: ChatGPT with Chain-of-Thought (CoT) Wei et al. (2022) prompting enhances the model's reasoning capabilities by encouraging it to generate intermediate reasoning steps before arriving at a final answer, improving performance on complex, multi-step problems.

- **LLMs with KG.** We compare our DAMR with fourteen LLMs with KG methods:

  - **UniKGQA**: UniKGQA Jiang et al. (2022) is a unified framework that integrates retrieval and reasoning for multi-hop question answering over knowledge graphs, combining subgraph retrieval, query decomposition, and neural reasoning in an end-to-end manner.

  - **DECAF**: DECAF Yu et al. (2022) is a joint framework for question answering over knowledge bases that simultaneously decodes logical forms and answers. By leveraging dual supervision, it enhances both symbolic reasoning accuracy and direct answer prediction in a unified architecture.

  - **KD-CoT**: KD-CoT Wang et al. (2023) is a framework that enhances the faithfulness of large language models by guiding Chain-of-Thought reasoning with external knowledge, improving accuracy in knowledge-intensive question answering tasks.

  - **Nutrea**: Nutrea Choi et al. (2023) proposes a neural tree search framework for context-guided multi-hop KGQA. It incrementally constructs reasoning trees by integrating question semantics and graph context, enabling efficient exploration of multi-hop paths for accurate answer prediction.

  - **ToG**: ToG Sun et al. (2023) is a framework that enables large language models to perform deep and responsible reasoning over knowledge graphs by combining structured graph information with iterative thinking and verification mechanisms for reliable multi-hop QA.

  - **RoG**: RoG Luo et al. (2023) is a framework that enhances the faithfulness and interpretability of large language model reasoning by grounding multi-hop question answering on knowledge graphs, integrating symbolic path tracking with natural language generation.

  - **KAPING**: KAPING Baek et al. (2023) introduces knowledge-augmented prompting by integrating structured triples into Chain-of-Thought (CoT) reasoning. It guides

| Methods | WebQSP-wiki | | MetaQA | | BioGraphletQA | |
|---|---|---|---|---|---|---|
| | Hits@1 | F1 | Hits@1 | F1 | Hits@1 | F1 |
| ChatGPT | 60.1 | 47.8 | 59.5 | 47.6 | 39.7 | 27.8 |
| FiDeLiS | 81.7 | 67.3 | 98.7 | 91.2 | 70.0 | 52.3 |
| DoG | 62.6 | 53.1 | 90.1 | **93.1** | 73.3 | 56.4 |
| DAMR | **88.2** | **72.5** | **99.2** | 87.9 | **80.9** | **60.7** |

Table 6: Performance comparison on WebQSP-wiki, MetaQA, and BioGraphletQA datasets.

| Methods | WebQSP | | CWQ | |
|---|---|---|---|---|
| | Hits@1 | F1 | Hits@1 | F1 |
| DAMR w/ history | 94.5 | 81.7 | 79.8 | 76.3 |
| DAMR | 94.0 | 81.7 | 78.0 | 75.1 |

Table 7: Performance of DAMR with and without historical reasoning traces on WebQSP and CWQ.

large language models to generate intermediate reasoning steps, enabling zero-shot multi-hop KGQA without task-specific fine-tuning.

– **ReasoningLM**: ReasoningLM Jiang et al. (2023) enhances pre-trained language models for KGQA by injecting subgraph structures into the input representation. It enables structural reasoning over retrieved subgraphs through a reasoning-aware encoder, improving performance on complex multi-hop queries.

– **FiDeLis**: FiDeLis Sui et al. (2024) proposes a faithfulness-aware KGQA framework that enhances reasoning consistency in LLMs by aligning generated logical forms with answer predictions. It introduces fidelity constraints to reduce hallucinations and improve answer correctness.

– **GNN-RAG**: GNN-RAG Mavromatis & Karypis (2024) integrates graph neural networks with retrieval-augmented generation by encoding knowledge subgraphs into LLMs' context. It enables structural reasoning over retrieved subgraphs, improving answer accuracy in KGQA through explicit graph-aware representations.

– **DoG**: DoG Ma et al. (2025a) is a flexible and reliable reasoning framework that enables large language models to generate and evaluate multiple reasoning paths over knowledge graphs through a debate-style process, enhancing robustness and answer faithfulness.

– **DuarL**: DuarL Liu et al. (2025) is a collaborative framework that integrates GNNs and LLMs for KGQA, where GNNs capture structural semantics and LLMs perform adaptive reasoning, enabling accurate and interpretable multi-hop QA.

– **DP**: DP Ma et al. (2025b) is a trustworthy reasoning framework that guides large language models using prior knowledge from knowledge graphs. It iteratively verifies and refines reasoning paths to enhance faithfulness, robustness, and answer accuracy in KGQA.

– **RwT**: RwT Shen et al. (2025) is a faithful KGQA framework that models multi-hop reasoning as tree-structured exploration over knowledge graphs, enabling large language models to generate interpretable reasoning paths and improve answer consistency and accuracy.

## D More Implementation Details.

In DAMR, the LLM-based planner operates entirely through external API calls without fine-tuning. It follows a fixed prompt template containing the input question and local outgoing relations, with a k-per-hop value of 3 and decoding temperature of 0.3. When no valid relation is produced, the planner outputs an empty list to terminate expansion. Qwen3-Embedding-8B is used to generate frozen embeddings for questions and relations. The path evaluation model is pre-trained on path pairs with a batch size of 8, a learning rate of 1e-5, and a 1:1 negative sampling ratio, then dynamically

| Method | WebQSP | | | CWQ | | |
|---|---|---|---|---|---|---|
| | Hits@1 | F1 | p-value | Hits@1 | F1 | p-value |
| FiDeLiS | $83.8 \pm 0.3$ | $77.9 \pm 0.2$ | $<0.01$ | $71.3 \pm 0.3$ | $64.1 \pm 0.2$ | $<0.01$ |
| DoG | $64.8 \pm 0.5$ | $54.7 \pm 0.3$ | $<0.01$ | $40.8 \pm 0.3$ | $46.3 \pm 0.3$ | $<0.01$ |
| **DAMR** | $\mathbf{93.9 \pm 0.2}$ | $\mathbf{81.5 \pm 0.3}$ | – | $\mathbf{77.8 \pm 0.2}$ | $\mathbf{74.9 \pm 0.3}$ | – |

Table 8: Performance (%) of DAMR and baselines on the WebQSP and CWQ datasets. Mean $\pm$ standard deviation and paired t-test p-values are reported.

| Methods | WebQSP | | CWQ | |
|---|---|---|---|---|
| | #Reasoning time | #GPU | #Reasoning time | #GPU |
| FiDeLis | 5.4 | 0 | 10.1 | 0 |
| DoG | 9.8 | 0 | 19.3 | 0 |
| DAMR | 4.3 | 0.6 | 7.6 | 0.8 |

Table 9: Average reasoning time (seconds) and GPU consumption on WebQSP and CWQ datasets.

fine-tuned during search using rollout-based pseudo-labels with a smaller learning rate of 5e-6 and batch size of 4. The hop limits are set to 3 for WebQSP and 4 for CWQ, and all experiments use fixed random seeds for reproducibility.

# E    MORE EXPERIMENTAL RESULTS

## E.1    MORE PERFORMANCE COMPARISON

To further evaluate the generality of DAMR beyond Freebase-based datasets, we conducted experiments on three additional KGQA benchmarks that differ in schema complexity and domain characteristics: MetaQA Ma et al. (2025a;b), BioGraphletQA[1], and WebQSP-Wiki Sun et al. (2023). MetaQA contains balanced subsets of 1-hop, 2-hop, and 3-hop questions, enabling a controlled analysis of multi-hop reasoning scalability. BioGraphletQA is a biomedical-domain benchmark designed to assess generalization to dense, domain-specific knowledge graphs. WebQSP-Wiki is a Wikidata-based reconstruction of WebQSP in which missing Freebase relations are explicitly labeled as PAD_RELATION, introducing realistic incompleteness and alias noise. As shown in Table 6, DAMR achieves state-of-the-art performance across all three datasets when compared with ChatGPT, FiDeLis Sui et al. (2024), and DoG Ma et al. (2025a), demonstrating strong generalization across diverse schemas, reasoning depths, and noise conditions.

To better understand how the planner benefits from feedback, we analyze its behavior from three perspectives. First, the LLM planner remains static to ensure stability and efficiency, since fine-tuning externally hosted APIs is impractical. Instead, DAMR decouples symbolic exploration (MCTS) from neural adaptation (path evaluation). Second, the planner implicitly benefits from the MCTS process, where plausible relations are reinforced through UCT updates and less effective ones are pruned over time. Third, to incorporate non-parametric feedback, we augment the planner's prompt with historical reasoning traces from previous rollouts. As shown in Table 7, this in-context adaptation consistently improves performance, indicating that accumulated reasoning experience helps the planner make more stable and informed decisions.

To verify the reliability of our findings, we performed additional statistical analyses by rerunning all experiments with five independent random seeds and conducting paired t-tests between DAMR and each baseline. This evaluation measures performance consistency across runs. The low variance and p-values below 0.01, reported in Table 8, confirm that DAMR's performance gains over FiDeLis Sui et al. (2024) and DoG Ma et al. (2025a) are statistically significant and not caused by random variation.

---

[1]https://zenodo.org/records/17381119

| Method | WebQSP | | CWQ | |
|---|---|---|---|---|
| | #Tokens | #Calls | #Tokens | #Calls |
| DAMR w/o PE | 5,037 | 8.9 | 13,870 | 22.9 |
| DAMR w/o FT | 4,929 | 8.3 | 12,432 | 21.5 |
| DAMR w/ GPT-4.1 | 7,742 | 13.8 | 23,773 | 42.3 |
| DAMR | **3,931** | **7.1** | **9,266** | **16.8** |

Table 10: Statistics of average number of LLM calls and token consumption per question on WebQSP and CWQ datasets.

| Method | WebQSP | | CWQ | |
|---|---|---|---|---|
| | Hits@1 | F1 | Hits@1 | F1 |
| DAMR w/o CT | 93.4 | 81.0 | 77.4 | 74.3 |
| DAMR w/o PE | 93.6 | 81.2 | 77.6 | 74.6 |
| DAMR w/o PR | 92.7 | 80.3 | 76.5 | 73.2 |
| DAMR | **94.0** | **81.7** | **78.0** | **75.1** |

Table 11: The results of ablation studies on the WebQSP and CWQ datasets. Bold results indicate the best performance.

## E.2 INFERENCE TIME AND MEMORY COMPARISON

To evaluate the practical runtime efficiency of DAMR, we compared its end-to-end reasoning latency and GPU usage with two representative baselines, FiDeLis Sui et al. (2024) and DoG Ma et al. (2025a), both of which also rely on asynchronous API calls to external LLMs. As shown in Table 9, DAMR achieves the lowest latency on both WebQSP and CWQ, demonstrating superior efficiency in API-dependent reasoning workflows. This improvement primarily results from restricting LLM invocations to the expansion phase and employing a lightweight Transformer-based path evaluator for local reasoning. During training and inference, DAMR requires only 0.6 GB and 0.8 GB of GPU memory on WebQSP and CWQ, respectively, while the baselines do not use GPU resources. CPU utilization for all models mainly comes from data preprocessing and handling asynchronous LLM calls.

## E.3 MORE ABLATION STUDY

To further assess the contribution of each module to computational efficiency, we measured the average number of tokens and API calls during reasoning, as shown in Table 10. The results indicate that both the path evaluator and the fine-tuning mechanism are critical for reducing LLM dependency and improving efficiency. Compared with directly evaluating reasoning paths using an LLM, DAMR significantly reduces token usage and API calls while maintaining high reasoning accuracy, demonstrating that its modular design effectively balances performance and computational cost.

To evaluate the sensitivity of DAMR to the design of its path scorer, we conducted ablation studies on three variants: DAMR w/o CT, which replaces the cross-attention mechanism with simple concatenation; DAMR w/o PE, which removes positional encodings; and DAMR w/o PR, which replaces the pairwise ranking loss with a binary cross-entropy loss. As shown in Table 11, DAMR achieves the best performance across both WebQSP and CWQ, demonstrating that each component contributes to its effectiveness. The removal of cross-attention or positional encodings causes moderate degradation, confirming their importance for modeling compositional semantics in multi-hop reasoning, while substituting the ranking loss leads to a larger drop, indicating that pairwise ranking better captures relative path plausibility.

## E.4 MORE SENSITIVITY ANALYSIS

To more thoroughly illustrate the impact of hyperparameter variations on model performance, we report detailed numerical results showing how performance fluctuates under different hyperparameter

| Method | WebQSP | | CWQ | |
|--------|--------|------|--------|------|
| | Hits@1 | F1 | Hits@1 | F1 |
| $k = 2$ | 93.0 | 80.9 | 76.6 | 73.8 |
| $k = 3$ | 94.0 | 81.7 | 78.0 | 75.1 |
| $k = 4$ | 94.0 | 81.8 | 77.8 | 75.0 |
| $k = 5$ | 93.9 | 81.7 | 78.0 | 75.2 |

Table 12: Hyperparameter sensitivity analysis of the number of selected relations $k$ on the WebQSP and CWQ datasets.

| Method | WebQSP | | CWQ | |
|--------|--------|------|--------|------|
| | Hits@1 | F1 | Hits@1 | F1 |
| $L = 2$ | 93.6 | 81.2 | 77.4 | 74.5 |
| $L = 3$ | 94.0 | 81.7 | 77.6 | 74.7 |
| $L = 4$ | 93.7 | 81.4 | 78.0 | 75.1 |
| $L = 5$ | 93.8 | 81.6 | 77.9 | 74.9 |

Table 13: Hyperparameter sensitivity analysis of the reasoning path length $L$ on the WebQSP and CWQ datasets.

settings. As presented in Table 12 and Table 13, these results provide a comprehensive understanding of the model's sensitivity and stability across a range of configurations.

### E.5 MORE THEORETICAL ANALYSIS

To empirically examine the stability and robustness of the online fine-tuning process in DAMR, we analyze the evolution of the path evaluator and its sensitivity to confidence margins. As shown in Figure 4, the predicted plausibility scores for positive and negative paths evolve steadily across refinement rounds, with positive-path scores (blue) increasing and negative-path scores (orange) decreasing, indicating stable convergence and clear separation without oscillation. To further evaluate robustness to noisy pseudo-pairs, we tested different confidence margins when forming training pairs. As presented in Table 14, performance remains consistent across margins ranging from 0.05 to 0.2, demonstrating that DAMR is largely insensitive to the threshold choice. Smaller margins slightly improve stability by filtering uncertain pairs, whereas larger margins remove too many samples and marginally reduce recall, confirming that the margin criterion effectively balances pseudo-label quality and robustness.

## F DISCUSSION

The design of DAMR is inspired by key limitations observed in previous KGQA frameworks. Existing LLM-based planners often entangle reasoning with generation, resulting in high computational overhead and limited adaptability. Static path scorers in prior graph reasoning models lack the ability to adjust to evolving exploration dynamics, while end-to-end reinforcement approaches frequently suffer from sparse and unstable rewards. DAMR overcomes these challenges through a decoupled and modular architecture that employs the LLM solely for semantic expansion, integrates a lightweight adaptive scorer for trajectory evaluation, and incorporates empirical-return-based feedback for stable self-correction. Together, these components enable DAMR to achieve both efficiency and adaptability, effectively bridging symbolic search and LLM-guided reasoning.

## G PROMPT TEMPLATE

We provide the prompt templates used by the LLM-based planner to select the top-$k$ most relevant relations from the candidate set at each step of path expansion in Figure 5, as part of the LLM Guided Path Expansion module.

| Methods | WebQSP | | CWQ | |
|---|---|---|---|---|
| | Hits@1 | F1 | Hits@1 | F1 |
| DAMR w margin=0.05 | 93.9 | 81.8 | 78.1 | 75.1 |
| DAMR w margin=0.1 | 93.9 | 81.7 | 77.9 | 75.1 |
| DAMR w margin=0.2 | 93.7 | 81.6 | 78.1 | 75.2 |
| DAMR | 94.0 | 81.7 | 78.0 | 75.1 |

Table 14: Performance of DAMR under different confidence margin settings on WebQSP and CWQ.

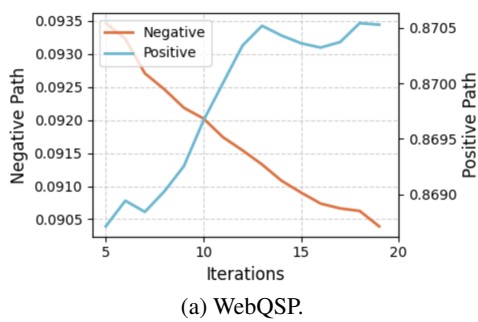
(a) WebQSP.

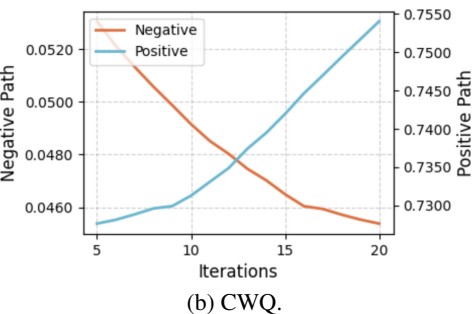
(b) CWQ.

Figure 4: Average predicted scores of the path evaluation model for real positive and negative paths during MCTS iterations on WebQSP and CWQ.

---

**Prompt Template for LLM-Guided Path Expansion**

**Role**
You are an expert assistant for Knowledge Graph Question Answering (KGQA). Your core capability is to deeply understand natural language questions and the semantics of knowledge graph relations to find the most relevant reasoning paths.

**Task**
Your task is to act as a **"Relation Retriever."** Given a natural language question and a list of candidate relations, you must analyze the semantics of the question and each relation to select up to `k` relations that are most likely to lead to the correct answer.

**Rules and Constraints**
- **Fidelity to Candidates**: Your selection of relations **MUST** come strictly from the provided `Candidate Relations` list. Do not invent or modify relations.
- **Quantity Limit**: Return no more than `k` relations. If multiple relations are highly relevant, order them from most to least relevant. If there are fewer than `k` relevant relations, return only those.
- **Output Format**: Your response **MUST** be a list of strings, containing the names of the relations you have selected.

**Example**
- **Input:**
    - `Question`: "who was the president after jfk died"
    - `Candidate Relations`: {"government.president", "government.president.successor", "location.location.containedby", "people.person.place_of_birth"}
    - `K`: 2
- **Output:**

    `["government.president", "government.president.successor"]`

**Your Task**
- `Question`: {question}
- `Candidate Relations`: {relations_list}
- `K`: {k}

**Output:**

`[]`

Figure 5: Prompt template used in the LLM-based planner to select top-k relations during reasoning.

## H  THE USE OF LARGE LANGUAGE MODELS (LLMS)

Large language models (LLMs) were only used to improve the clarity, grammar, and fluency of the manuscript. They were not involved in the development of research ideas, experimental design, data analysis, or any other aspect of the scientific content.

