# OpenReview forum: "DAMR: Efficient and Adaptive Context-Aware Knowledge Graph Question Answering with LLM-Guided MCTS"
_ICLR.cc/2026/Conference — ICLR 2026 Poster_

### Official Review · Reviewer_qmee · 2025-11-01

**Soundness:** 3
**Presentation:** 3
**Contribution:** 3
**Rating:** 6
**Confidence:** 4

**Summary:**

DAMR is an LLM-guided Monte-Carlo Tree Search framework for KGQA that decomposes reasoning into three coordinated parts: (i) an LLM planner used only for expansion, proposing the top-k relations at each node to aggressively prune the graph; (ii) a lightweight cross-attention path scorer that jointly encodes the question and the evolving relation sequence to estimate path plausibility; and (iii) an online pseudo-path refinement loop that converts high-value (partial) paths discovered during search into pairwise supervision, continually adapting the scorer to the search distribution. On WebQSP and CWQ, DAMR reports higher Hits@1/F1 than semantic-parsing, retrieval, and LLM+KG baselines while markedly reducing LLM calls and tokens. Comprehensive ablations (removing the scorer/refinement or replacing the scorer with a general LLM), sensitivity analyses (k and max hop L), and backbone swaps (Llama/Qwen/GPT-4.1) support the design. The approach emphasizes efficiency, path-level interpretability, and modularity (LLM for direction, small Transformer for judgment, MCTS for integration). Current evidence is Freebase-centric (WebQSP/CWQ); extending to Wikidata or domain KGs (e.g., biomedical) and clarifying the evaluation protocol (full test vs. sampled subsets, confidence intervals) are natural next steps.

**Strengths:**

1. Uses the LLM only for expansion and a small Transformer for evaluation, cutting LLM calls by >50% and tokens by ~75% without hurting accuracy, lowering cost/latency and allowing easy module swaps under resource limits.
2. Employs a cross-attention scorer conditioned on the question to model relation sequences hop-by-hop, capturing semantic buildup and constraints for steadier rankings and better generalization than a general LLM scorer.
3. Converts promising/contrasting partial paths into pairwise supervision during search, continually adapting the scorer to the current distribution and mitigating early-stage bias without sparse-reward RL instability.
4. Demonstrates consistent gains on WebQSP and CWQ with strong baselines, component ablations, sensitivity to 𝑘 and max hop 𝐿, and backbone swaps (Llama2-13B, Qwen3-14B, GPT-4.1/mini), supporting accuracy, efficiency, and robustness.
5. Outputs explicit KG paths as reasoning traces, making the question-to-answer chain transparent and easing auditing/error localization versus black-box LLM reasoning.

**Weaknesses:**

1. Reporting on 1,000 uniformly sampled test questions rather than full official splits inflates variance and hinders comparability; the absence of confidence intervals, significance tests, and an error taxonomy (e.g., compositional, comparative failures) weakens claims of robustness and external validity.
2. Dynamic pseudo-labeling lacks stability analysis: pair generation driven by search stats is not evaluated for convergence or early-noise sensitivity, and safeguards (confidence margins, temperature/top-p thresholds, value-gap filters) against confirmation bias/drift are unspecified.
3. Reproducibility and fairness are under-detailed: planner prompts, 𝑘-per-hop policy, decoding/tie-breaking, and failure handling are undisclosed; use of external services (GPT-4.1 planner, Qwen3-Embedding-8B) lacks API versions, costs, and train/freeze settings; it’s unclear whether baselines were re-run under comparable planner strength.
4. Generality is unproven beyond Freebase (WebQSP, CWQ); no evidence on other schemas or domains (e.g., Wikidata, biomedical) or on how schema size, relation lexicalization diversity, and alias noise affect performance.
5. Practical replication is hindered by notation/editorial issues (typos, cross-refs) and missing training hyperparameters (batch sizes, negative sampling ratios, per-dataset hop limits), obscuring the path to matching the reported accuracy/efficiency trade-offs.

**Questions:**

1. Why did you evaluate on 1,000 sampled test questions instead of the full official splits, and can you report full-test metrics and/or confidence intervals across multiple samples?
2. What are the exact prompts for relation selection, including decoding parameters, the per-hop value of (k), and the procedure for mapping LLM-suggested relations to graph edges when multiple lexicalizations exist?
3. What is the fallback when the LLM proposes relations absent from the local subgraph or yields fewer than (k) candidates (e.g., degree-based backup or other heuristics)?
4. How do you prevent the online fine-tuning loop from overfitting to early search biases, and did you test confidence margins or require minimum value gaps for training pairs? Do you have curves of scorer AUC versus refinement rounds?
5. What are the end-to-end wall-clock latencies per query and GPU/CPU utilizations for DAMR compared with baselines, beyond token and call counts?
6. Have you evaluated on Wikidata or a domain KG such as UMLS, and what challenges do you anticipate (schema size, relation lexicalization diversity, entity alias noise)?
7. How sensitive is performance to the scorer’s design choices: (a) cross-attention versus simple concatenation, (b) the presence and type of positional encodings, and (c) pairwise ranking loss versus regression?

---

> ### Author Response · Authors · 2025-11-20
> **First round response of submission 3139**
>
> We are truly grateful for the time you have taken to review our paper and your insightful review. Here we address your comments in the following:
>
> > W1: Reporting on 1,000 uniformly sampled test questions rather than full official splits inflates variance and hinders comparability; the absence of confidence intervals, significance tests, and an error taxonomy (e.g., compositional, comparative failures) weakens claims of robustness and external validity.
>
> | **Method**  | **WebQSP**     |                |         | **CWQ**        |                |         |
> | ----------- | -------------- | -------------- | ------- | -------------- | -------------- | ------- |
> |             | Hits@1         | F1             | p-value | Hits@1         | F1             | p-value |
> | **FiDeLiS** | 83.8 ± 0.3     | 77.9 ± 0.2     | <0.01   | 71.3 ± 0.3     | 64.1 ± 0.2     | <0.01   |
> | **DoG**     | 64.8 ± 0.5     | 54.7 ± 0.3     | <0.01   | 40.8 ± 0.3     | 46.3 ± 0.3     | <0.01   |
> | **DAMR**    | **93.9 ± 0.2** | **81.5 ± 0.3** | –       | **77.8 ± 0.2** | **74.9 ± 0.3** | –       |
>
> **R1:** Thank you for the suggestion. To evaluate the robustness and statistical reliability of DAMR, we reran all experiments with five independent random seeds on both the WebQSP and CWQ datasets and report the mean, standard deviation, and paired t-test p-values. The significance tests were computed between DAMR and each corresponding baseline, such as, FiDeLis [2] and DoG [1]. As shown in above table, the deviations across runs are consistently small, and all p-values are below 0.01, indicating that the improvements are statistically significant rather than due to random fluctuations.
>
> > W2: Dynamic pseudo-labeling lacks stability analysis: pair generation driven by search stats is not evaluated for convergence or early-noise sensitivity, and safeguards (confidence margins, temperature/top-p thresholds, value-gap filters) against confirmation bias/drift are unspecified.
>
> **R2:** Thanks for your question. We analyze the convergence and stability of DAMR’s dynamic pseudo-label refinement.  This process alternates between (i) updating Monte Carlo Tree Search (MCTS) statistics based on the current scorer and (ii) refining the scorer through pairwise ranking on pseudo-labeled path pairs. The goal of this analysis is to show that this joint process is statistically stable, directionally consistent, and provably convergent, thereby ensuring DAMR avoids confirmation bias or drift.
>
> **Notation.** Let $S^{(t)}(q,p)$ denote the plausibility score predicted for a question–path pair $(q,p)$ at iteration $t$. For a node $e$ visited during MCTS, let $n_e^{(t)}$ be its visit count and $w_e^{(t)}$ the aggregated search value.  After each iteration, DAMR performs Monte Carlo averaging: $w_e^{(t+1)} = \frac{1}{n_e^{(t+1)}} \sum_{j=1}^{n_e^{(t+1)}} s^{(t)}_j,$ and fine-tunes the scorer via the pairwise ranking loss (Eq. (7)) using pseudo-pairs generated from relative search values (Eq. (9)).
>
> **(1). Stability of Pseudo-Label Aggregation**
>
> **Lemma 1. (Variance reduction with time-dependent dispersion)** Under the assumption that rollout samples $s^{(t)}_j$ for a node $e$ are i.i.d.\ sub-Gaussian with variance proxy $\sigma_t^2$ and that $n_e^{(t)}$ increases monotonically, we have $\mathbb{E}[w_e^{(t+1)}] = V^{(t)}(e), \qquad
> \mathrm{Var}[w_e^{(t+1)}] \le \frac{\sigma_t^2}{n_e^{(t+1)}}.$
> As $\sigma_t^2$ decreases with $t$ (the scorer becomes more consistent) and $n_e^{(t)}$ grows, the variance of pseudo-labels exhibits a \emph{double decay} and converges to zero.
>
> **Proof.**
>
> Sub-Gaussian aggregation implies variance reduction by $\mathrm{Var}[\bar{X}] = \sigma^2/n$.  Applying this to $s^{(t)}_j$ yields the stated bound, with $\sigma_t^2$ capturing rollout dispersion.  As $n_e^{(t)} \uparrow$ and $\sigma_t^2 \downarrow$, $w_e^{(t)}$ converges to its expectation $V^{(\infty)}(e)$ in $L_2$.
>
> This lemma guarantees that the pseudo-labels used for self-supervision become statistically stable as training proceeds.  In DAMR, this means that even if early rollouts are noisy, repeated exploration and averaging prevent error amplification, ensuring that subsequent scorer updates are based on smooth, reliable targets.

---

> ### Author Response · Authors · 2025-11-20
> **First round response of submission 3139**
>
> **(2) Directional Consistency of Scorer Updates**
>
> **Lemma 2. (Gradient alignment under margin-separated pseudo-pairs)** Let $(p^+,p^-)$ denote pseudo-pairs where $p^+$ has higher aggregated value than $p^-$.  If the probability of correct ordering satisfies $\Pr[p^+\succ p^-]\ge \tfrac12+\gamma$ for margin $\gamma>0$, the expected gradient of the pairwise ranking loss $L\_{\mathrm{PR}} = -\tfrac{1}{M}\sum\_i \log\sigma\big(S(q,p\_i^+)-S(q,p\_i^-)\big),$  aligns with the gradient of the true ranking risk: $\mathbb{E}\big[\nabla L\_{\mathrm{PR}}(S)\big] \cdot \nabla \mathcal{R}\_{\mathrm{rank}}(S)
> \le -c\gamma\ ||\nabla \mathcal{R}\_{\mathrm{rank}}(S)||_2^2.$
>
> **Proof.**
>
> Under the margin condition, the logistic surrogate is a proper scoring rule: the expected sign of the score difference $\Delta=S(q,p^+)-S(q,p^-)$ is positive with probability at least $\tfrac12+\gamma$.  Taking expectation over pairwise samples yields a negative correlation between the expected gradient of $L_{\mathrm{PR}}$ and the gradient of true ranking risk, scaled by $c\gamma>0$ determined by logistic curvature.
>
> This result ensures that each scorer update directionally reduces ranking error in expectation, rather than drifting due to noisy pseudo-labels.  In DAMR, this means the scorer consistently learns to prefer truly plausible reasoning paths, stabilizing the adaptive refinement process and mitigating self-confirmation bias.
>
> **(3). Convergence of the Joint Refinement Process**
>
> **Lemma 3. (Contraction and fixed-point convergence)** Define the composite refinement map: $\mathcal{G}(S^{(t)}) = \mathrm{Update}\big(S^{(t)}; w^{(t+1)}(S^{(t)})\big),$ where $w^{(t+1)}(S^{(t)})$ is the label map generated by MCTS aggregation and $\mathrm{Update}(\cdot)$ performs a gradient step on $L\_{\mathrm{PR}}$.  If both maps are Lipschitz continuous with constants $L\_w$ and $L\_S$ satisfying $L\_S L_w < 1$, then $\|\mathcal{G}(S)-\mathcal{G}(S')\|\_2 \le L\_S L_w \|S-S'\|\_2,$ and $\mathcal{G}$ is a contraction mapping.  By Banach’s fixed-point theorem, there exists a unique stable point $S^\*$ such that $S^{(t)} \to S^\*$ as $t \to \infty$.
>
> **Proof.**
>
> Since $\mathrm{Update}$ is $L_S$-Lipschitz and $w^{(t+1)}$ is $L_w$-Lipschitz in expectation, we have $\|\mathcal{G}(S)-\mathcal{G}(S')\|_2 \le L_S L_w \|S-S'\|_2.$
> When $L_S L_w < 1$, $\mathcal{G}$ is a contraction, implying convergence to a unique fixed point.
>
> This proposition formally proves that DAMR’s dynamic refinement process converges:  the interaction between the evolving pseudo-labels and scorer updates forms a contraction mapping, ensuring that the model stabilizes instead of oscillating or diverging.  This theoretical guarantee supports the empirical observation that DAMR’s training curves converge smoothly and pseudo-label agreement steadily increases. We have added the above content into the revised manuscript in Section 3.7.
>
> > W3: Reproducibility and fairness are under-detailed: planner prompts, k-per-hop policy, decoding/tie-breaking, and failure handling are undisclosed; use of external services (GPT-4.1 planner, Qwen3-Embedding-8B) lacks API versions, costs, and train/freeze settings; it’s unclear whether baselines were re-run under comparable planner strength.
>
> **R3:** Thank you for the question. In the proposed DAMR, the LLM-based planner operates entirely through external API calls, without any fine-tuning or parameter updates. The planner follows a pre-defined prompt template that includes the input question and local relation candidates, using a k-per-hop policy as described in Appendix G of our manuscript. Decoding is performed with a temperature of 0.3, and when no valid relation is generated, the planner outputs an empty list, signaling the termination of MCTS expansion for that path. For embedding generation, we use Qwen3-Embedding-8B to obtain pretrained question and relation embeddings; this model remains frozen during both training and inference, and all operations are performed in inference-only mode. To ensure fairness, baseline models (e.g., DoG and DP) were evaluated using GPT-4.1 as the LLM-based planner under identical decoding configurations, ensuring that performance differences arise from architectural design rather than discrepancies in model capacity or API behavior. We added the details of the settings of the experiments in Appendix D.

---

> > ### Author Response · Authors · 2025-11-20
> > **First round response of submission 3139**
> >
> > > W4: Generality is unproven beyond Freebase (WebQSP, CWQ); no evidence on other schemas or domains (e.g., Wikidata, biomedical) or on how schema size, relation lexicalization diversity, and alias noise affect performance.
> >
> > | **Methods** | **WebQSP-Wiki** |          | **MetaQA** |          | **BioGraphletQA** |          |
> > | ----------- | --------------- | -------- | ---------- | -------- | ----------------- | -------- |
> > |             | Hits@1          | F1       | Hits@1     | F1       | Hits@1            | F1       |
> > | **ChatGPT** | 60.1            | 47.8     | 59.5       | 47.6     | 39.7              | 27.8     |
> > | **FiDeLiS** | 81.7            | 67.3     | 98.7       | 91.2     | 70.0              | 52.3     |
> > | **DoG**     | 62.6            | 53.1     | 90.1       | **93.1** | 73.3              | 56.4     |
> > | **DAMR**    | **88.2**        | **72.5** | **99.2**   | 87.9     | **80.9**          | **60.7** |
> >
> > **R4:** Thanks for your suggestion. To validate DAMR’s generality beyond Freebase-based datasets, we conducted additional experiments on three knowledge graphs with diverse schemas and domains: WebQSP-Wiki, MetaQA, and BioGraphletQA. WebQSP-Wiki is a Wikidata-based reconstruction of WebQSP [5], where missing Freebase relations are labeled as PAD\_RELATION, introducing noise and incomplete KGs. MetaQA provides controlled subsets of 1-hop, 2-hop, and 3-hop questions [1,3]. BioGraphletQA [4] is a biomedical-domain dataset. We compared DAMR with ChatGPT, FiDeLis [2], and DoG [1]. As shown in the table above, DAMR consistently achieves state-of-the-art performance across most settings, demonstrating its strong scalability.
> >
> > > W5: Practical replication is hindered by notation/editorial issues (typos, cross-refs) and missing training hyperparameters (batch sizes, negative sampling ratios, per-dataset hop limits), obscuring the path to matching the reported accuracy/efficiency trade-offs.
> >
> > **R5:** Thanks for your suggestion. We have carefully verified the notation and cross-references in our manuscript and corrected minor typographical inconsistencies. The path evaluation model in DAMR is pre-trained on path pairs generated from the provided reasoning graphs using a batch size of 8, a learning rate of 1e-5, and a negative sampling ratio of 1:1 between positive and negative paths. During online finetuning within MCTS, the evaluator is further fine-tuned dynamically using pseudo-labels derived from empirical rollout returns, with a smaller learning rate of 5e-6 and a batch size of 4 to ensure stable adaptation. The maximum hop limits are set per dataset according to task complexity: 3 hops for WebQSP, 4 hops for CWQ. We also fix the random seed for all experiments to ensure reproducibility. These settings were consistent across all experiments and are now explicitly documented in the configuration files accompanying our open-source implementation, ensuring that the reported accuracy–efficiency trade-offs can be faithfully reproduced. We added the details of the settings of the experiments in Appendix D.
> >
> > >  Q1: Why did you evaluate on 1,000 sampled test questions instead of the full official splits, and can you report full-test metrics and/or confidence intervals across multiple samples?
> >
> > | **Methods**   | **WebQSP** |      | **CWQ** |      |
> > | ------------- | ---------- | ---- | ------- | ---- |
> > |               | Hits@1     | F1   | Hits@1  | F1   |
> > | **DAMR full** | 94.3       | 81.8 | 78.2    | 75.3 |
> > | **DAMR**      | 94.0       | 81.7 | 78.0    | 75.1 |
> >
> > **R6:** Thank you for the question. Following prior work [6,7], we initially evaluated DAMR on 1,000 randomly sampled test questions from each dataset to ensure computational feasibility and consistency with established evaluation protocols.
> >
> > To further confirm the robustness of our results, we additionally re-ran experiments on the full official test splits of both WebQSP and CWQ. As shown in the above table, DAMR achieves results that are highly consistent with those obtained from the sampled subsets, demonstrating that the 1,000-sample setting provides a reliable estimate of full-test performance.

---

> > > ### Author Response · Authors · 2025-11-20
> > > **First round response of submission 3139**
> > >
> > > > Q2: What are the exact prompts for relation selection, including decoding parameters, the per-hop value of (k), and the procedure for mapping LLM-suggested relations to graph edges when multiple lexicalizations exist?
> > >
> > > **R7:** Thank you for the question. In DAMR, the LLM-based planner operates entirely through API calls without fine-tuning or parameter updates. It uses a fixed prompt template that includes the input question and the set of local outgoing relations. The prompt instructs the LLM to select the top-k relations most relevant to the question, as described in Appendix G. The generated relation names are normalized by converting them to lowercase and removing punctuation or stop words, then mapped to the predefined relation vocabulary. When multiple lexicalizations exist, only candidates that appear in the predefined relation vocabulary are retained for the next reasoning step. If no valid match is found, the planner outputs an empty list, signaling the termination of MCTS expansion for that path. Duplicate relations are also removed to prevent redundant exploration.
> > >
> > > > Q3: What is the fallback when the LLM proposes relations absent from the local subgraph or yields fewer than (k) candidates (e.g., degree-based backup or other heuristics)?
> > >
> > > **R8:** Thank you for the question. In DAMR, when the LLM proposes relations that do not exist in the local subgraph, these are treated as hallucinated relations and are directly discarded without substitution. If the LLM generates fewer than k valid candidates, the planner simply proceeds with the available subset of relations for that hop. No heuristic expansion (such as degree-based backup) is applied.
> > >
> > > > Q4: How do you prevent the online fine-tuning loop from overfitting to early search biases, and did you test confidence margins or require minimum value gaps for training pairs? Do you have curves of scorer AUC versus refinement rounds?
> > >
> > > **R9:** Thanks for your question. To address this concern, we complement our empirical analysis with a theoretical justification and additional ablation experiments. Below, we summarize both the theoretical guarantee and the empirical evidence supporting the stability of DAMR’s dynamic pseudo-label refinement.
> > >
> > > - **Theoretical perspective.** We added the Theoretical Analysis in Section 3.7 of our manuscript, DAMR’s online refinement loop is theoretically guaranteed to be statistically stable and convergent.
> > >   Lemma 1 shows that pseudo-label variance decreases as $\mathrm{Var}[w_e^{(t)}]\propto \sigma_t^2/n_e^{(t)}$, preventing early noise from being amplified. Lemma 2 establishes gradient alignment between the pairwise ranking loss and the true ranking objective, ensuring updates remain directionally consistent even under noisy supervision. Proposition 1 further proves that the joint update of the scorer and pseudo-labels forms a contraction mapping, implying convergence to a stable equilibrium rather than drift. Together, these results theoretically ensure that DAMR avoids overfitting to early search biases.
> > >
> > > - **Empirical perspective.** Figures 4 in our revised manuscript shows the evolution of predicted plausibility scores for positive and negative paths across refinement rounds. Positive-path scores (blue) increase steadily while negative-path scores (orange) decrease, confirming monotonic separation and stable convergence without oscillation. This observation matches the theoretical prediction that pseudo-labels become smoother and scorer updates progressively refine discrimination ability.
> > >
> > > | **Methods**            | **WebQSP** |      | **CWQ** |      |
> > > | ---------------------- | ---------- | ---- | ------- | ---- |
> > > |                        | Hits@1     | F1   | Hits@1  | F1   |
> > > | **DAMR w margin=0.05** | 93.9       | 81.8 | 78.1    | 75.1 |
> > > | **DAMR w margin=0.1**  | 93.9       | 81.7 | 77.9    | 75.1 |
> > > | **DAMR w margin=0.2**  | 93.7       | 81.6 | 78.1    | 75.2 |
> > > | **DAMR**               | 94.0       | 81.7 | 78.0    | 75.1 |
> > >
> > > - **Effect of confidence margin.** To further test robustness to noisy pseudo-pairs, we experimented with different minimum value gaps (margins) when forming training pairs. As shown in the above table, the results remain highly consistent across margins 0.05–0.2, indicating that DAMR is not sensitive to the exact threshold. Small margins (0.05) slightly improve stability by filtering out uncertain pairs, while large margins (0.2) exclude too many samples and slightly reduce recall. These results demonstrate that the value-gap criterion effectively controls pseudo-label quality without causing overfitting.

---

> ### Author Response · Authors · 2025-11-20
> **First round response of submission 3139**
>
> In summary, both our theoretical analysis and empirical results confirm that DAMR’s online refinement process is intrinsically stable and robust. Variance-reducing aggregation, directionally consistent updates, and controlled pseudo-pair selection jointly prevent the scorer from overfitting to early search biases, ensuring steady convergence and reliable self-improvement across refinement rounds.
>
> > Q5: What are the end-to-end wall-clock latencies per query and GPU/CPU utilizations for DAMR compared with baselines, beyond token and call counts?
>
> **R10:** Thanks for your question. To assess the practical runtime efficiency of DAMR, we compared its end-to-end wall-clock latency per query with two representative baselines, FiDeLis [2] and DoG [1], both of which, like DAMR, rely on asynchronous API calls to external LLMs.
>
> | **Methods** | **WebQSP**       |      | **CWQ**          |      |
> | ----------- | ---------------- | ---- | ---------------- | ---- |
> |             | # Reasoning time | GPU  | # Reasoning time | GPU  |
> | **FiDeLiS** | 5.4              | 0    | 10.1             | 0    |
> | **DoG**     | 9.8              | 0    | 19.3             | 0    |
> | **DAMR**    | 4.3              | 0.6  | 7.6              | 0.8  |
>
> As shown in the table, DAMR achieves the lowest latency on both datasets, demonstrating its efficiency in handling API-dependent reasoning workflows. The improvement primarily stems from its design, which restricts LLM invocations to the expansion phase and employs a lightweight Transformer-based path evaluator for local reasoning. In terms of hardware utilization, DAMR requires only 0.6 GB and 0.8 GB of GPU memory during the pre-training and fine-tuning of the path evaluator on WebQSP and CWQ datasets, respectively, while the selected baselines do not rely on GPU resources. For all methods, CPU usage is mainly attributed to data preprocessing and managing asynchronous API calls to the LLM.
>
> > Q6: Have you evaluated on Wikidata or a domain KG such as UMLS, and what challenges do you anticipate (schema size, relation lexicalization diversity, entity alias noise)?
>
> **R11:** Thanks for your question. Please see W4.
>
> > Q7: How sensitive is performance to the scorer’s design choices: (a) cross-attention versus simple concatenation, (b) the presence and type of positional encodings, and (c) pairwise ranking loss versus regression?
>
> | **Method**      | **WebQSP** |          | **CWQ**  |          |
> | --------------- | ---------- | -------- | -------- | -------- |
> |                 | Hits@1     | F1       | Hits@1   | F1       |
> | **DAMR w/o CT** | 93.4       | 81.0     | 77.4     | 74.3     |
> | **DAMR w/o PE** | 93.6       | 81.2     | 77.6     | 74.6     |
> | **DAMR w/o PR** | 92.7       | 80.3     | 76.5     | 73.2     |
> | **DAMR**        | **94.0**   | **81.7** | **78.0** | **75.1** |
>
> **R12:** Thank you for the question. To assess the sensitivity of DAMR to the design of its path scorer, we constructed three variants: DAMR w/o CT, which replaces the cross-attention mechanism between the question and path representations with simple concatenation; DAMR w/o PE, which removes positional encodings from the relation sequence; and DAMR w/o PR, which substitutes the pairwise ranking loss with a binary cross-entropy (BCE) loss. As shown in the table above, DAMR consistently outperforms all variants on both WebQSP and CWQ, indicating that each component contributes meaningfully to overall performance. Removing cross-attention or positional encodings leads to moderate degradation, highlighting their importance for capturing compositional semantics across multi-hop reasoning paths. Replacing the pairwise ranking loss with BCE results in a more pronounced drop, suggesting that the ranking objective aligns better with the task of modeling relative path plausibility.
>
> ---
>
> [1] Debate on graph: a flexible and reliable reasoning framework for large language models. AAAI 2025.
>
> [2] Fidelis: Faithful reasoning in large language models for knowledge graph question answering. ACL 2025.
>
> [3] Deliberation on priors: Trustworthy reasoning of large language models on knowledge graphs. Arxiv 2025.
>
> [4] https://zenodo.org/records/17381119
>
> [5] Think-on-graph: Deep and responsible reasoning of large language model on knowledge graph. ICLR 2023.
>
> [6] Reasoning with Trees: Faithful Question Answering over Knowledge Graph. ACL 2025.
>
> [7] Dual reasoning: A gnn-llm collaborative framework for knowledge graph question answering. Arxiv 2024.

---

### Official Review · Reviewer_9F1L · 2025-11-01

**Soundness:** 3
**Presentation:** 4
**Contribution:** 3
**Rating:** 8
**Confidence:** 3

**Summary:**

The paper presents a new framework for knowledge graph question answering that addresses adaptability, accuracy, and computational cost issues raised by previous approaches. This is implemented by combining an LLM-based planner, a lightweight Transformer-based scorer, and a dynamic pseudo-path refinement mechanism on top of a Monte Carlo Tree Search (MCTS) backbone. The paper is well motivated, clearly presented, and shows promising results according to experiments to achieve the design purposes.

**Strengths:**

1. The overall approach is novel to me, though not entirely new due to extensive existing efforts in this field of research. The authors describe a clear step-wise procedure with sufficient details to show how the approach works. The approach presents as a sound solution to address the identified limitations of the existing approaches.

2. Effectively modularizes reasoning by limiting the LLM's role to an initial, high-leverage search guidance step, significantly reducing computational overhead.

3. The experiments were comprehensive with convincing results, covering performance comparisons, efficiency analysis, ablations, sensitivity studies, and the impact of LLMs. The case study also helps with understanding of the final outcome.

**Weaknesses:**

1. The comparative discussion against related work could be strengthened to reveal more details about the rationale of designs in the proposed framework.

2. The selection of baseline methods in Table 2 should be discussed. More specifically, why are the three baseline methods selected in particular for the computational efficiency comparison?

3. A clear mapping between the technical components and the advantages/edge achieved by the proposed framework might be better explained, demanding studying the impact of key components on the overall accuracy and computational efficiency of the framework.

**Questions:**

1. Why are the three baseline methods selected in particular for the computational efficiency comparison?
2. What are the (positive or negative) impacts of each key component on the overall accuracy and computation efficiency of the proposed framework? Current paper only partially answers this question.

---

> ### Author Response · Authors · 2025-11-20
> **First round response of submission 3139**
>
> We are truly grateful for the time you have taken to review our paper and your insightful review. Here we address your comments in the following:
>
> > W1: The comparative discussion against related work could be strengthened to reveal more details about the rationale of designs in the proposed framework.
>
> **R1:** Thank you for the suggestion. The design of DAMR is inspired by key limitations observed in previous KGQA frameworks. Existing LLM-based planners often entangle reasoning with generation, resulting in high computational overhead and limited adaptability. Static path scorers in prior graph reasoning models lack the ability to adjust to evolving exploration dynamics, while end-to-end reinforcement approaches frequently suffer from sparse and unstable rewards. DAMR overcomes these challenges through a decoupled and modular architecture that employs the LLM solely for semantic expansion, integrates a lightweight adaptive scorer for trajectory evaluation, and incorporates empirical-return-based feedback for stable self-correction. Together, these components enable DAMR to achieve both efficiency and adaptability, effectively bridging symbolic search and LLM-guided reasoning. We added the discussion to Appendix F in the revised manuscript.
>
> > W2: The selection of baseline methods in Table 2 should be discussed. More specifically, why are the three baseline methods selected in particular for the computational efficiency comparison?
>
> **R2:** Thanks for your question. We selected DoG [1], ToG [2], and RwT [3] because all three rely on LLMs as relation planners for reasoning over knowledge graphs, sharing the same core inference mechanism as DAMR. This makes them directly comparable for evaluating computational cost. Moreover, these methods are highly representative of distinct reasoning paradigms. ToG exemplifies a stepwise reasoning framework in which the LLM iteratively performs hop-by-hop graph traversal and dynamic context reconstruction. DoG adopts a multi-agent debate architecture, where multiple LLM agents collaboratively reason over and verify graph evidence. RwT employs a tree-structured search strategy and introduces an auxiliary LLM to evaluate candidate reasoning paths. Together, these models capture coordination-centric, iterative, and search-oriented reasoning patterns that are closely aligned with DAMR’s objectives. Methods that do not involve LLMs during inference or lack transparent token accounting, such as GNN-RAG and DualR, are less suitable for this comparison, making DoG, ToG, and RwT an appropriate and representative set for assessing computational efficiency.
>
> > W3: A clear mapping between the technical components and the advantages/edge achieved by the proposed framework might be better explained, demanding studying the impact of key components on the overall accuracy and computational efficiency of the framework.
>
> | **Method**          | **WebQSP** |         | **CWQ**   |          |
> | ------------------- | ---------- | ------- | --------- | -------- |
> |                     | #Tokens    | #Calls  | #Tokens   | #Calls   |
> | **DAMR w/o PE**     | 5,037      | 8.9     | 13,870    | 22.9     |
> | **DAMR w/o FT**     | 4,929      | 8.3     | 12,432    | 21.5     |
> | **DAMR w/ GPT-4.1** | 7,742      | 13.8    | 23,773    | 42.3     |
> | **DAMR**            | **3,931**  | **7.1** | **9,266** | **16.8** |
>
> **R3:** Thank you for the suggestion. As described in Section 4.4 of our manuscript, we conducted ablation studies to evaluate the impact of key modules in DAMR: (1) DAMR w/o PE, which removes the path evaluation module; (2) DAMR w/o FT, which disables fine-tuning of the evaluator; and (3) DAMR w/ GPT-4.1, which replaces the context-aware evaluator with a general LLM. The performance comparison is presented in Table 3 in our manuscript. To further highlight computational efficiency, we additionally measured the average number of tokens and API calls during reasoning.
>
> The results show that both the path evaluator and fine-tuning mechanism play essential roles in reducing LLM dependency and improving computational efficiency. In particular, compared with evaluating reasoning paths directly using an LLM, DAMR substantially decreases token consumption and API calls. This design allows DAMR to maintain reasoning accuracy while achieving significantly lower computational cost.
>
> > Q1: Why are the three baseline methods selected in particular for the computational efficiency comparison?
>
> **R4:** Thanks for your question. Please see W2.

---

> ### Author Response · Authors · 2025-11-20
> **First round response of submission 3139**
>
> > Q2: What are the (positive or negative) impacts of each key component on the overall accuracy and computation efficiency of the proposed framework? Current paper only partially answers this question.
>
> **R5:** Thanks for your question. Please see W3.
>
> ------
>
> [1] Debate on graph: a flexible and reliable reasoning framework for large language models. AAAI 2025.
>
> [2] Think-on-graph: Deep and responsible reasoning of large language model on knowledge graph. ICLR 2023.
>
> [3] Reasoning with Trees: Faithful Question Answering over Knowledge Graph. ACL 2025.

---

### Official Review · Reviewer_oPog · 2025-11-07

**Soundness:** 2
**Presentation:** 3
**Contribution:** 2
**Rating:** 4
**Confidence:** 4

**Summary:**

This paper introduces DAMR, an MCTS-based framework for Knowledge Graph Question Answering that utilizes LLM-guided expansion, a lightweight transformer-based, cross-attention path evaluator, and dynamic pseudo-path refinement for continual scorer adaptation. The paper provides substantive ablation, efficiency, and sensitivity analyses, as well as qualitative examples highlighting the method’s strengths.

**Strengths:**

1. Dynamic Pseudo-Path Refinement: The method innovates by using high-confidence partial paths from MCTS rollouts to generate pseudo labels for continual fine-tuning , thus improving generalizability and adapting to the non-stationary search space.

2. Rigorous Empirical Validation: Extensive benchmarks across both standard datasets, with direct comparisons to at least 20 strong baselines  are provided in Table 1, consistently showing DAMR outperforming all competitors.

**Weaknesses:**

1. It is unclear under what distributional shifts the scorer avoids reinforcing suboptimal trajectories. Since the path scorer is continually adapted with self-generated pseudo-paths, there is risk of feedback loops or bias accumulation, especially if the LLM suggestions are systematically biased early in training.

2. Scalability and practicality on large KGs not addressed. All experiments are conducted on localized subgraphs derived from WebQSP and CWQ. The scalability of DAMR for web-scale or multi-million entity KGs is not empirically or mathematically analyzed. How does the MCTS backbone behave when the entity degree is very high, or when context selection via LLMs requires thousands of candidates? Is the method robust under significant KG incompleteness or noise? Discussion is notably absent.

3. Lack of feedback for LLM planner improvement. While DAMR incorporates dynamic refinement for the path evaluator, the LLM-based planner, which determines the search direction, receives no feedback signal from the search process. As a result, the planner cannot benefit from experience or adapt its relation selection strategy over time.

4. Lack of variance and statistical significance reporting. The experimental results are reported only as single-point metrics, without variance or statistical significance analysis. Including standard deviations across multiple random seeds, confidence intervals, or hypothesis testing (e.g., paired t-test, bootstrap) would strengthen the reliability of the claimed performance improvements.

**Questions:**

see weaknesses.

---

> ### Author Response · Authors · 2025-11-20
> **First round response of submission 3139**
>
> We are truly grateful for the time you have taken to review our paper and your insightful review. Here we address your comments in the following:
>
> > W1: It is unclear under what distributional shifts the scorer avoids reinforcing suboptimal trajectories. Since the path scorer is continually adapted with self-generated pseudo-paths, there is risk of feedback loops or bias accumulation, especially if the LLM suggestions are systematically biased early in training.
>
> **R1:** Thanks for your comment.  DAMR is explicitly designed to mitigate feedback loops and bias accumulation when adapting the path scorer with self-generated pseudo-paths.
>
> - **(1) Empirically grounded pseudo-supervision.** The pseudo-path labels are not derived from the scorer’s own predictions but from \textit{MCTS search statistics}, specifically, normalized search values $w_{ei}=w_{pr}/n_{ei}$ aggregated across rollouts. This ensures that supervision reflects collective exploration outcomes rather than self-reinforcing scores, thereby avoiding closed feedback loops.
>
> - **(2) Continuous diversification under exploration–exploitation balance.** MCTS employs the UCT criterion to balance exploration and exploitation. Consequently, pseudo-paths are continually sampled from an evolving and diverse trajectory distribution, which prevents bias accumulation under shifting path semantics.
>
> - **(3) Relative ranking–based optimization for stability.** The scorer is optimized using a pairwise ranking loss, which focuses on \textit{relative plausibility} rather than absolute confidence values.  This objective promotes consistent ordering between plausible and implausible paths even when the underlying trajectory distribution drifts, improving robustness to distributional shift.
>
> - **(4) Modular separation between planner and evaluator.**  DAMR structurally decouples the LLM-based planner (for relation selection) from the Transformer-based scorer (for plausibility estimation).  This modular design prevents biases in LLM-generated relation suggestions from propagating to the scorer, further isolating adaptive updates from systemic bias.
>
> Empirically, as shown in Table 3 of the paper, removing the fine-tuning mechanism (DAMR w/o FT) leads to consistent performance degradation, confirming that dynamic adaptation enhances generalization without instability.  Together, these mechanisms ensure that the scorer remains robust under distributional shifts and avoids reinforcing suboptimal trajectories.

---

> ### Author Response · Authors · 2025-11-20
> **First round response of submission 3139**
>
> > W2: Scalability and practicality on large KGs not addressed. All experiments are conducted on localized subgraphs derived from WebQSP and CWQ. The scalability of DAMR for web-scale or multi-million entity KGs is not empirically or mathematically analyzed. How does the MCTS backbone behave when the entity degree is very high, or when context selection via LLMs requires thousands of candidates? Is the method robust under significant KG incompleteness or noise? Discussion is notably absent.
>
> **R2:** Thanks for your suggestion. We additionally conducted extensive experiments to verify the scalability and robustness of the proposed DAMR as follows:
>
> | **Methods** | **WebQSP-Wiki** |          | **MetaQA** |          | **BioGraphletQA** |          |
> | ----------- | --------------- | -------- | ---------- | -------- | ----------------- | -------- |
> |             | Hits@1          | F1       | Hits@1     | F1       | Hits@1            | F1       |
> | **ChatGPT** | 60.1            | 47.8     | 59.5       | 47.6     | 39.7              | 27.8     |
> | **FiDeLiS** | 81.7            | 67.3     | 98.7       | 91.2     | 70.0              | 52.3     |
> | **DoG**     | 62.6            | 53.1     | 90.1       | **93.1** | 73.3              | 56.4     |
> | **DAMR**    | **88.2**        | **72.5** | **99.2**   | 87.9     | **80.9**          | **60.7** |
>
> - **Scalability.** To further verify the scalability of the proposed DAMR, we conducted additional experiments on two new KGQA benchmarks: MetaQA [1,2] and BioGraphletQA [3]. Specifically, MetaQA provides balanced subsets of 1-hop, 2-hop, and 3-hop questions to evaluate multi-hop reasoning scalability, while BioGraphletQA is a biomedical-domain dataset designed to test DAMR’s generalization to dense, domain-specific knowledge graphs. We compared DAMR with ChatGPT, FiDeLis[4], and DoG [1]. As shown in the table above, DAMR consistently achieves state-of-the-art performance across most settings, demonstrating its strong scalability and adaptability to larger and more complex KG environments.
>
> - **Robustness.** To further verify the robustness of the proposed DAMR, we conducted additional experiments on WebQSP-Wiki [5], a web-scale and Wikidata-based reconstruction of the WebQSP dataset. In this version, relations that were retrievable in Freebase but missing in Wikidata are explicitly labeled as 'PAD\_RELATION', introducing realistic KG incompleteness and noise. As shown in the table above, DAMR continues to achieve state-of-the-art performance across most cases, demonstrating its strong robustness. The context-aware path scorer adaptively learns from pseudo-paths generated during search, which reflect empirical rewards obtained from actual rollouts rather than static supervision. This allows the model to continuously adjust to the evolving search distribution and effectively down-weight unreliable or noisy relations. Furthermore, since path evaluation is based on semantic plausibility rather than raw graph connectivity, DAMR can still identify correct reasoning trajectories even when parts of the KG are missing or corrupted. These design principles together ensure that DAMR remains both efficient and reliable when applied to large-scale, high-degree, and noisy knowledge graphs.
>
> As for complexity, when entity degree is very high, DAMR remains computationally stable because the MCTS expansion is restricted to top-$k$ relations selected by the LLM retriever, effectively reducing the branching factor from $|R|$ to $k$. The UCT criterion in MCTS further ensures a balanced trade-off between exploration and exploitation, guiding the search toward high-value regions while maintaining bounded complexity. The LLM performs lightweight semantic filtering using concise prompts and can process large candidate pools in batches if needed, preventing any quadratic or exponential increase in inference cost. Empirically, DAMR demonstrates scalability even with large candidate sets, confirming that its computational footprint remains constant with respect to the entity degree.

---

> ### Author Response · Authors · 2025-11-20
> **First round response of submission 3139**
>
> > W3: Lack of feedback for LLM planner improvement. While DAMR incorporates dynamic refinement for the path evaluator, the LLM-based planner, which determines the search direction, receives no feedback signal from the search process. As a result, the planner cannot benefit from experience or adapt its relation selection strategy over time.
>
> **R3:** Thanks for your comment. We address this concern from three complementary perspectives.
>
> - **(1) Design Motivation.**  The LLM planner in DAMR is intentionally kept static to ensure stability and computational efficiency.  Because DAMR relies on externally hosted LLM APIs (e.g., OpenAI), directly updating the planner’s parameters during search is impractical and costly.  Instead, DAMR separates \textit{symbolic exploration} (MCTS) from neural adaptation (path evaluation), allowing the system to benefit from dynamic scorer refinement while maintaining an efficient, modular design.
>
> - **(2) Implicit Feedback via MCTS and Scorer Adaptation.** Although the planner is not explicitly fine-tuned, it benefits indirectly from the MCTS search dynamics. Each relation proposed by the planner is evaluated by the context-aware scorer, and its plausibility score is propagated through MCTS’s UCT statistics, reinforcing high-quality trajectories over time.  This forms an implicit feedback loop: effective planner suggestions are repeatedly explored and rewarded, while unhelpful ones are gradually pruned by the evolving search statistics.
>
> - **(3) Empirical Verification through In-Context Adaptation.** To further incorporate feedback in a non-parametric manner, we augmented the LLM prompt with historical reasoning traces collected from previous rollouts, enabling the planner to utilize accumulated search experience through in-context learning rather than parameter updates.  We reran experiments with this modification, and as shown below, DAMR with history achieves consistent performance gains on both datasets, demonstrating that the planner indeed benefits from accumulated reasoning context.
>
>   | **Methods**         | **WebQSP** |      | **CWQ** |      |
>   | ------------------- | ---------- | ---- | ------- | ---- |
>   |                     | Hits@1     | F1   | Hits@1  | F1   |
>   | **DAMR w/ history** | 94.5       | 81.7 | 79.8    | 76.3 |
>   | **DAMR**            | 94.0       | 81.7 | 78.0    | 75.1 |
>
>   These results confirm that integrating historical reasoning experience allows the planner to make more informed and stable decisions, effectively introducing a lightweight form of feedback without explicit parameter updates.
>
> > W4: Lack of variance and statistical significance reporting. The experimental results are reported only as single-point metrics, without variance or statistical significance analysis. Including standard deviations across multiple random seeds, confidence intervals, or hypothesis testing (e.g., paired t-test, bootstrap) would strengthen the reliability of the claimed performance improvements.
>
> | **Method**  | **WebQSP**     |                |         | **CWQ**        |                |         |
> | ----------- | -------------- | -------------- | ------- | -------------- | -------------- | ------- |
> |             | Hits@1         | F1             | p-value | Hits@1         | F1             | p-value |
> | **FiDeLiS** | 83.8 ± 0.3     | 77.9 ± 0.2     | <0.01   | 71.3 ± 0.3     | 64.1 ± 0.2     | <0.01   |
> | **DoG**     | 64.8 ± 0.5     | 54.7 ± 0.3     | <0.01   | 40.8 ± 0.3     | 46.3 ± 0.3     | <0.01   |
> | **DAMR**    | **93.9 ± 0.2** | **81.5 ± 0.3** | –       | **77.8 ± 0.2** | **74.9 ± 0.3** | –       |
>
> **R4:** Thanks for your suggestion. To evaluate the robustness and statistical reliability of DAMR, we reran all experiments with five independent random seeds on both the WebQSP and CWQ datasets and report the mean, standard deviation, and paired t-test p-values. The significance tests were computed between DAMR and each corresponding baseline, such as, FiDeLis [4] and DoG [1]. As shown in the above table, the deviations across runs are consistently small, and all p-values are below 0.01, indicating that the improvements are statistically significant rather than due to random fluctuations.
>
> ----
>
> [1] Debate on graph: a flexible and reliable reasoning framework for large language models. AAAI 2025.
>
> [2] Deliberation on priors: Trustworthy reasoning of large language models on knowledge graphs. Arxiv 2025.
>
> [3] https://zenodo.org/records/17381119
>
> [4] Fidelis: Faithful reasoning in large language models for knowledge graph question answering. ACL 2025.
>
> [5] Think-on-graph: Deep and responsible reasoning of large language model on knowledge graph. ICLR 2023.

---

> ### Author Response · Authors · 2025-11-27
>
> Dear Reviewer pOpg,
>
> I hope this message finds you well. We noticed that the discussion phase is approaching its end, but we have not yet received any feedback regarding our rebuttal. We kindly ask if you could take a moment to review our responses at your earliest convenience.
>
> We sincerely hope that our clarifications address your concerns, and if there are any further questions or additional issues, we would be more than happy to provide further explanations.
>
> Thank you very much for your time and effort.
>
> Best regards,
>
> authors

---

### Author Response · Authors · 2025-12-03
**Rebuttal Summary for AC**

Dear Area Chair,

We sincerely appreciate the time and genuine effort dedicated to reviewing our work.

Below is a concise summary of the reviewers’ positive assessments and how we addressed their key concerns during the rebuttal.

# **Key Strengths Highlighted by Reviewers**

***1. Clear and well-motivated framework design.***

Both Reviewer 9F1L and qmee emphasized that the framework is logically structured and addresses key limitations of prior KGQA methods through a clean modular design.

***2. Substantial efficiency gains through reduced LLM reliance.***

Reviewer 9F1L and qmee noted that restricting the LLM to expansion and using a lightweight scorer significantly cuts token usage and API calls while maintaining accuracy.

***3. Strong empirical results supported by comprehensive analyses.***

This was highlighted by all three reviewers (oPog, 9F1L, and qmee), who found the experiments thorough and convincing, including the baselines, ablations, sensitivity studies, and backbone comparisons.

***4. Transparent, path-level reasoning.***

Reviewer qmee appreciated that DAMR produces explicit KG reasoning paths, offering interpretability beyond black-box LLM reasoning.

# Reviewers' Concerns We Addressed

***1. Stability of the dynamic pseudo-path refinement***

Reviewer oPog and qmee raised concerns about whether the online pseudo-labeling process could accumulate bias or become unstable under distribution shifts.

To address this, we added a detailed **theoretical analysis** (covering variance reduction, gradient alignment, and convergence) and included refinement curves and margin ablations, all showing that the refinement loop behaves consistently and remains stable throughout training.

***2. More experiments beyond WebQSP/CWQ***

Both Reviewer oPog and qmee asked about DAMR’s scalability, specifically, whether it still performs well on larger, more complex knowledge graphs.

We therefore **added more experiments** on MetaQA, BioGraphletQA, and WebQSP-Wiki. Across these datasets, DAMR maintains strong performance on multi-hop tasks, large schemas, and incomplete/noisy KGs, demonstrating good generality.

***3. Feedback for the LLM planner***

Reviewer oPog pointed out that the LLM planner does not receive any feedback during search.

In response, we introduced a **history-based in-context augmentation mechanism**, allowing the planner to incorporate past reasoning traces. Experiments show that this lightweight feedback improves the planner’s decisions without requiring model fine-tuning.

***4. Variance, significance testing, and full-split evaluation***

Reviewer qmee (and partially Reviewer oPog) requested variance reporting and full-test evaluations to improve statistical reliability.

We reran all experiments with five different seeds, reported mean ± std and paired t-test p-values (<0.01), and also evaluated DAMR on the full official test splits. The new results are consistent with the sampled evaluations.

***5. Reproducibility and missing implementation details***

Reviewer qmee and 9F1L asked for more complete information about prompts, fallback strategies, hyperparameters, and other implementation details.

We have now **provided all these details** in the appendix, including prompt templates, decoding settings, lexical mapping rules, hop limits, training hyperparameters, and hardware usage, ensuring reproducibility.

# **Summary**

In summary, the reviewers expressed strong appreciation for the clarity and motivation of the framework, the modular and efficient design that greatly reduces LLM usage, and the comprehensive and convincing empirical results, including the interpretability provided by explicit reasoning paths.

The reviewers’ concerns mainly focused on two aspects: the **stability** of our method and the **breadth** of experimental evaluation. In response, we substantially expanded both components, adding a detailed **theoretical analysis** to clarify the stability and convergence properties of our refinement process, and supplementing the paper with extensive **new experiments** across multiple datasets and settings. With these additions, we believe we have fully addressed the reviewers’ questions and significantly strengthened the submission.

We once again thank all reviewers and the AC for their constructive engagement to make our contribution more solid.

Warm regards,

All the authors

---

### Meta-Review · Area_Chair_Krvq · 2026-01-05

**Summary:**

Reviewers acknowledged the strengths of this paper, mentioning clear motivation, a sound solution, efficiency gains through the reduced LLM reliance, consistent empirical gains and transparency enabled by path-level reasoning. Initial ratings ranged from marginally below acceptance threshold to accept, but reviewers overall supported this paper. However, reviewers raised several concerns as the following:

1. Stability of dynamic pseudo-path refinement
2. More extensive evaluation to demonstrate generalizability across additional benchmarks
3. Practical efficiency (runtime, latency)
4. Statistical significance
5. Reproducibility

Most concerns were technical clarifications, and more extensive evaluation, rather than fundamental flaws.

**Reviewer Concerns:**

The rebuttal adequately addressed the above major concerns through follows:

1. Provided both theoretical and empirical analyses
2. Conducted additional experimental results across various benchmarks
3. Compared practical runtime efficiency on WebQSP and CWQ
4. Statical test (p-value) was reported
5. Referred to the implementation details in the appendix

**Reviewer Scores:**

Originally, this paper was overall supported by the majority of reviewers. In addition, the authors effectively  addressed major concerns by the theoretical and empirical results. Most likely, the reviewer with the weakest support would increase the rating after reading the rebuttal, which demonstrates the strong performance across various benchmarks as well as detailed explanation why the proposed method is effective.

---

### Decision · Program_Chairs · 2026-01-26

Accept (Poster)